# ABSTOPK: RETHINKING SPARSE AUTOENCODERS FOR BIDIRECTIONAL FEATURES

**Xudong Zhu, Mohammad Mahdi Khalili, Zhihui Zhu**
Department of Computer Science & Engineering
The Ohio State University
{zhu.3944, khalili.17, zhu.3440}@osu.edu

## ABSTRACT

Sparse autoencoders (SAEs) have emerged as powerful techniques for interpretability of large language models (LLMs), aiming to decompose hidden states into meaningful semantic features. While several SAE variants have been proposed, there remains no principled framework to derive SAEs from the original dictionary learning formulation. In this work, we introduce such a framework by unrolling the proximal gradient method for sparse coding. We show that a single-step update naturally recovers common SAE variants, including ReLU, JumpReLU, and TopK. Through this lens, we reveal a fundamental limitation of existing SAEs: their sparsity-inducing regularizers enforce non-negativity, preventing a single feature from representing bidirectional concepts (e.g., male vs. female). This structural constraint fragments semantic axes into separate, redundant features, limiting representational completeness. To address this issue, we propose AbsTopK SAE, a new variant derived from the $\ell_0$ sparsity constraint that applies hard thresholding over the largest-magnitude activations. By preserving both positive and negative activations, AbsTopK uncovers richer, bidirectional conceptual representations. Comprehensive experiments across multiple LLMs and seven probing and steering tasks show that AbsTopK improves reconstruction fidelity, enhances interpretability, and enables single features to encode contrasting concepts. Remarkably, AbsTopK matches or even surpasses the Difference-in-Mean method—a supervised approach that requires labeled data for each concept and has been shown in prior work to outperform SAEs.

## 1 INTRODUCTION

The pursuit of interpretability has become a central objective in modern machine learning, as it is essential for the assurance, debugging, and fine-grained control of large language models (LLMs) (Marks et al., 2025; Park et al., 2023; Luo et al., 2024; Arora et al., 2018). Within this domain, sparse dictionary learning methods (Poggio & Serre, 2006; Fel et al., 2023), and specifically sparse autoencoders (SAEs), have re-emerged as a prominent methodology for systematically enumerating the latent concepts a model may employ in its predictions (Hindupur et al., 2025; Bussmann et al., 2024; Rajamanoharan et al., 2025; Gao et al., 2025).

An SAE decomposes a model's hidden representations into an overcomplete basis of latent features (Elhage et al., 2022; Thasarathan et al., 2025), which ideally correspond to abstract, data-driven concepts whose linear superposition reconstructs the original activation vector (Higgins et al., 2017; Fel, 2025). Empirical evidence indicates that SAE latents capture semantically coherent features across diverse domains. In LLMs, these features exhibit selectivity for specific entities (e.g., *Golden Gate Bridge*), linguistic behaviors (e.g., sycophantic phrasing), and symbolic systems (e.g., Hebrew script) (Templeton et al., 2024; Csordás et al., 2024; Durmus et al., 2024). Similarly, in vision models, they respond to distinct objects (e.g., barbers, dog shadows) and complex scene properties (e.g., foreground-background separation, facial detection in crowds) (Fel, 2024; Thasarathan et al., 2025). In protein models, they have been shown to correlate with functional elements such as binding sites and structural motifs (Garcia & Ansuini, 2025; Adams et al., 2025). The discovery of such interpretable, semantically grounded features suggests a natural avenue for steering models: by amplifying, suppressing, or combining specific latents, one can intervene to modulate downstream

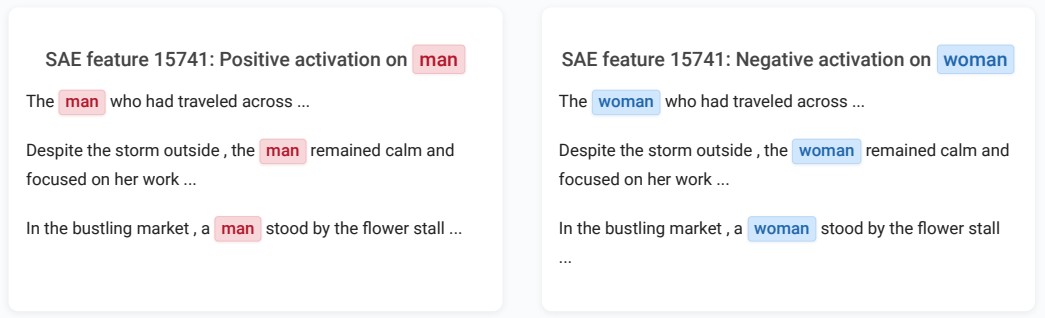

Figure 1: **AbsTopK enables single latent features to encode opposing concepts by leveraging both positive and negative activations.** To test this, we generated controlled sentence pairs with only one differing token (*man* vs. *woman*). The shown feature activates positively for *man* and negatively for *woman*, demonstrating bidirectional encoding. Unlike conventional SAEs, which are restricted by a non-negativity constraint, AbsTopK more compactly captures opposing semantics within a single dimension, yielding richer and more coherent representations.

behavior, which is a principal motivation for research into SAEs (Gao et al., 2025; Bricken et al., 2023; Kantamneni et al., 2025). This control is predicated on the assumption that the concepts identified by SAEs faithfully correspond to the features underlying a model's predictions (Arditi et al., 2024; Uppaal et al., 2024; Engels et al., 2025).

However, recent studies suggest that simpler supervised techniques such as Difference-in-Means (DiM) can outperform SAEs on practical steering benchmarks and tasks (Arditi et al., 2024; Wu et al., 2025). Unlike SAEs, which are unsupervised and can simultaneously identify multiple latent features, DiM requires labeled data and is typically limited to extracting a single vector for a pre-specified concept. Nevertheless, these findings raise questions regarding the degree to which SAEs recover a model's internal features. The fact that comparatively simple baselines can rival or even surpass SAEs on downstream control tasks suggests that the features identified by SAEs may only partially align with the model's underlying neural representations, thereby casting doubt on their fidelity as faithful explanatory tools.

We posit that one source of this misalignment lies in a structural limitation of SAEs recently proposed for studying LLMs—including the vanilla version with ReLU (Cunningham et al., 2023), the JumpReLU variant (Rajamanoharan et al., 2025), and the TopK variant (Gao et al., 2025): their systematic neglect of negative activations, despite evidence that many meaningful directions in representation space are inherently bidirectional (Mao et al., 2022). The *linear representation hypothesis* (Mikolov et al., 2013) suggests that a model's internal states can be approximated as linear combinations of semantic vectors, where conceptual transformations correspond to both positive and negative displacements along these vector axes (Arora et al., 2018; Uppaal et al., 2024; Luo et al., 2024). The DiM approach builds on this assumption, requiring labeled datasets that capture both sides of a concept, with positive and negative examples defining a bidirectional semantic axis. Classic word analogies, such as the vector operation $v_{\text{king}} - v_{\text{man}} + v_{\text{woman}} \approx v_{\text{queen}}$ (Pennington et al., 2014), illustrate how semantic differences are encoded as generalizable vector offsets. Nevertheless, by enforcing non-negativity or retaining only the TopK activations (Bussmann et al., 2024; Gao et al., 2025), conventional SAEs either fragment such contrastive concepts into separate, unidirectional bases (e.g., "male" and "female") or discard one direction of the semantic axis entirely. This not only undermines the representational capacity of SAEs but also limits their usefulness for controlled interventions, where traversing both directions of a semantic axis is often essential. This raises the following questions: *is the use of nonnegative activations truly essential for the success of SAEs, or does it instead constrain their ability to capture richer representations? More concretely, can SAEs be improved by allowing negative activations, thereby enabling the discovery of bidirectional concepts?*

**Contributions.** In this work, we address these questions by $(i)$ introducing a unified framework for designing SAEs, $(ii)$ proposing a new variant, AbsTopK SAE, and $(iii)$ conducting comprehensive experiments across four LLMs and seven probing and steering tasks to demonstrate that

allowing negative activations further enhances SAEs, yielding improved reconstruction fidelity and greater interpretability.

Our contributions are summarized as follows:

- **A Unified Framework for Designing SAEs.** We introduce a principled framework for designing SAEs by unrolling the proximal gradient method for sparse coding with sparsity-inducing regularizers. A single-step update naturally induces common SAE variants, including ReLU, JumpReLU, and TopK (Templeton et al., 2024; Gao et al., 2025; Rajamanoharan et al., 2025). This framework provides a rigorous tool for analyzing their implicit regularizers and identifying shared limitations.
- **Absolute TopK (AbsTopK) for Learning Bidirectional Features** Building on this framework, we propose a new SAE variant, termed absolute TopK (AbsTopK) derived from the vanilla sparsity constraint ($\ell_0$ norm) without a non-negative constraint, which results in a hard-thresholding operator that selects the largest-magnitude activations. By preserving both positive and negative activations, AbsTopK SAE allows a single feature to capture opposing concepts (Figure 1), thereby uncovering richer bidirectional representations.
- **Comprehensive Empirical Validation.** We conduct a comprehensive empirical evaluation across four LLMs, comparing the proposed AbsTopK SAE with TopK and JumpReLU SAEs on a suite of seven probing and steering tasks, along with three unsupervised metrics. The results demonstrate that AbsTopK outperforms TopK and JumpReLU SAEs, producing representations with higher fidelity and interpretability. Additionally, a case study illustrates that AbsTopK can encode a bidirectional semantic axis within a single latent feature, effectively capturing contrasting concepts. Notably, AbsTopK achieves performance comparable to—or even exceeding—the Difference-in-Mean method, which relies on labeled data and has been shown in prior work to outperform SAEs.

## 2 FROM PROXIMAL INTERPRETATIONS OF SAEs TO ABSTOPK

### 2.1 PRELIMINARIES

We denote vectors by lowercase bold letters (e.g., $\boldsymbol{x}$) and matrices by uppercase bold letters (e.g., $\boldsymbol{X}$). With an input sequence of $N$ tokens, $\boldsymbol{X} = \{\boldsymbol{x}_1, \ldots, \boldsymbol{x}_N\}$, where each $\boldsymbol{x}_j$ denotes the embedding of the $j$-th token, the LLM can be viewed as a function $f : \mathbb{R}^{d \times N} \to \mathbb{R}^{V \times N}$, where $V$ is the vocabulary size and $f(\boldsymbol{X})$ gives the output logits for all tokens in the sequence. For our purposes, we abstract away the internal details of $f$ and instead study the representations in the hidden layers. Consider interventions at layer $\ell$ in the residual stream. Supposing that the model comprises $L$ layers, then $f$ can be decomposed as

$$f(\boldsymbol{X}) = \phi_{\ell+1:L}\big(\phi_{1:\ell}(\boldsymbol{X})\big), \tag{1}$$

where $\phi_{1:\ell}(\boldsymbol{X})$ denotes the representation after the first $\ell$ layers and $\phi_{\ell+1:L}$ represents the remaining computation from layer $\ell$ to $L$. We denote by $\boldsymbol{x}_j^{(\ell)}$ the embedding of the residual stream at the $\ell$-th layer corresponding to the $j$-th token of the input sequence $\boldsymbol{X}$. In the following presentation, when the context is clear, we omit the superscript ($\ell$) and the subscript (token index $j$) for notational simplicity, and denote the hidden embedding of a token in a given layer by $\boldsymbol{x}$.

The linear representation hypothesis (Park et al., 2023) assumes that the hidden representation $\boldsymbol{x}$ can be expressed as a linear superposition of latent concepts:

$$\boldsymbol{x} = \sum_{p=1}^{P} \alpha_p \boldsymbol{h}_p + \text{residual}, \tag{2}$$

where $\{\boldsymbol{h}_p\}_{p=1}^{P}$ are referred to as concept directions or feature vectors, such as gender or sentiment, $\{\alpha_p\}$ are the corresponding coefficients, and residual term captures approximation error as well as context-specific variation that is not explained by the selected concepts. Since a particular token—although it encodes information from previous tokens in the context—typically contains only a small subset of concepts or features, its representation is expected to be *sparse*; that is, most of the coefficients $\alpha_p$ are zero, resulting in a *sparse linear representation*, often simply referred to as a *sparse representation*. Importantly, the coefficients $\alpha_p$ are not required to be non-negative. In fact, for binary concepts, the sign of a coefficient is semantically meaningful, indicating opposite

directions; for example, it distinguishes whether a contextually appropriate token should be "king" or "queen" (when the context involves a monarch) (Park et al., 2023).

We note that, although (2) is written in terms of one-dimensional concept directions $\boldsymbol{h}_p$, some concepts may in practice be better modeled as low-dimensional feature subspaces. As discussed in (Engels et al., 2025), such multi-dimensional features can still be captured by SAEs, and our use of the linear representation hypothesis is compatible with this view.

To find these concept directions or feature vectors, supervised approaches such as the Difference-in-Mean (DiM) method construct labeled datasets for each target attribute. While effective for isolating specific concepts, these methods are inherently limited to predefined features and do not scale to the large number of latent dimensions present in LLM representations. In contrast, dictionary learning provides an unsupervised and scalable alternative: it can simultaneously recover a more complete dictionary that approximates the underlying concept directions, uncovering a richer and more comprehensive set of latent features than DiM, which is typically restricted to a single concept vector. Consequently, while DiM may achieve stronger control on a specific concept (Wu et al., 2025), dictionary-learning methods have gained popularity due to their ability to uncover a richer, more comprehensive set of latent features.

## 2.2 DICTIONARY LEARNING AND THE PROXIMAL PERSPECTIVE ON SPARSE AUTOENCODERS

In a nutshell, dictionary learning (Olshausen & Field, 1996) seeks to construct a dictionary $\boldsymbol{D}$ consisting of basis vectors $\{\boldsymbol{d}_1, \ldots, \boldsymbol{d}_P\}$, which are also called as *atoms*, such that it can approximately provide sparse linear combination for all token embeddings $\boldsymbol{x}$ from the same layer. Since the total number of concept vectors $P'$ is unknown, $P$ is typically set to a relatively large value to ensure that as many concepts as possible can be learned. This typically requires solving a training problem of form (Mairal et al., 2011)

$$\min_{\boldsymbol{D} \in \mathbb{R}^{d \times P}, \boldsymbol{b} \in \mathbb{R}^d} \mathbb{E}_{\boldsymbol{x}} \left[ \min_{\boldsymbol{z} \in \mathbb{R}^P} \underbrace{\frac{1}{2} \|\boldsymbol{x} - (\boldsymbol{D}\boldsymbol{z} + \boldsymbol{b})\|_2^2}_{g(\boldsymbol{z})} + \lambda R(\boldsymbol{z}) \right], \tag{3}$$

where $R(\boldsymbol{z})$ is a sparsity-inducing regularizer, $\lambda > 0$ controls the trade-off between reconstruction fidelity and sparsity, $\boldsymbol{b}$ is an additional bias vector. In classical dictionary learning, the data is often preprocessed to have zero global mean, so the bias term is not used. Alternatively, the bias term can be incorporated into the dictionary as $\boldsymbol{D}\boldsymbol{z} + \boldsymbol{b} = [\boldsymbol{D} \quad \boldsymbol{b}] \begin{bmatrix} \boldsymbol{z} \\ 1 \end{bmatrix}$. In this work, however, we explicitly include $\boldsymbol{b}$ to align with the structure of commonly used SAEs which will be described later.

The main challenge in solving the problem (3) lies in jointly estimating both the dictionary $(\boldsymbol{D}, \boldsymbol{b})$ and the sparse coefficients $\boldsymbol{z}$. When one of these variables is fixed, optimizing over the other becomes relatively easier,[1] though still nontrivial in practice. In particular, given a dictionary $\boldsymbol{D}$ and bias $\boldsymbol{b}$, the problem reduces to finding a sparse approximation of $\boldsymbol{x}$, a step commonly referred to as *sparse coding*. An efficient method for solving this problem is the proximal gradient method (Parikh et al., 2014; Silva & Rodriguez, 2020), which is especially suitable when the regularizer $R(\boldsymbol{z})$ is non-differentiable, such as the $\ell_1$ norm used in Lasso (Tibshirani, 1996) or $\ell_0$ norm that directly enforce sparsity (Foucart, 2011; Bao et al., 2014; Rajamanoharan et al., 2025).

**Proximal gradient methods induce encoders** For a function $r : \mathbb{R}^d \to \mathbb{R}$, its proximal operator is defined by (Parikh et al., 2014)

$$\text{prox}_r(\boldsymbol{u}) = \arg\min_{\boldsymbol{v} \in \mathbb{R}^d} \frac{1}{2} \|\boldsymbol{v} - \boldsymbol{u}\|^2 + r(\boldsymbol{v}).$$

---

[1]This observation has motivated alternating minimization methods such as MOD (Cai et al., 2016) and K-SVD (Aharon et al., 2006).

Now starting from an initialization $\boldsymbol{z}^{(0)}$, the proximal gradient method for optimizing $\boldsymbol{z}$ in (3) performs iterative updates of the form

$$\boldsymbol{z}^{(t+1)} = \text{prox}_{\mu\lambda R}\left(\boldsymbol{z}^{(t)} - \mu\nabla g(\boldsymbol{z}^{(t)})\right) = \text{prox}_{\mu\lambda R}\left(\boldsymbol{z}^{(t)} - \mu\boldsymbol{D}^\top(\boldsymbol{D}\boldsymbol{z}^{(t)} + \boldsymbol{b} - \boldsymbol{x}))\right), \quad (4)$$

where $\mu > 0$ is the step size. This perspective naturally leads to *unrolled networks* (Gregor & LeCun, 2010; Chen et al., 2022), where each proximal gradient step can be interpreted as a layer in a neural network that iteratively refines the latent code $\boldsymbol{z}$ while enforcing sparsity (Daubechies et al., 2004). In particular, with $\boldsymbol{z}^{(0)} = \boldsymbol{0}$ and $\mu = 1$, the first update becomes

$$\boldsymbol{z}^{(1)} = \text{prox}_{\lambda R}\left(\boldsymbol{D}^\top\boldsymbol{x} - \boldsymbol{D}^\top\boldsymbol{b}\right). \quad (5)$$

Since a single proximal gradient step yields only an approximate solution, inspired by prior work on unrolled networks, we replace the fixed parameters $\boldsymbol{D}$ and $\boldsymbol{b}$ with learnable counterparts: a trainable weight matrix $\boldsymbol{W}$ in place of $\boldsymbol{D}$, and a learnable bias vector $\boldsymbol{b}_\text{e}$ in place of $-\boldsymbol{D}^\top\boldsymbol{b}$, thereby yielding a more accurate approximation to the sparse coding solution. Then the update (5) becomes

$$\boldsymbol{z}^{(1)} = \text{prox}_{\lambda R}\left(\boldsymbol{W}^\top\boldsymbol{x} + \boldsymbol{b}_\text{e}\right), \quad (6)$$

which resembles an encoder. The following result shows that certain regularizers give rise to proximal operators commonly used in SAEs.

**Lemma 1.** *Denote by* $\text{ReLU}_\lambda, \text{JumpReLU}_\theta, \text{TopK}_k$ *as the following operators:*

$$(\text{ReLU}_\lambda(\boldsymbol{u}))_i = \max\{u_i - \lambda, 0\}, \quad (\text{JumpReLU}_\theta(\boldsymbol{u}))_i = \begin{cases} 0, & u_i < \theta, \\ u_i, & u_i \geq \theta, \end{cases}$$

$$(\text{TopK}_k(\boldsymbol{u}))_i = \begin{cases} \max\{u_i, 0\}, & i \in \mathcal{T}_k(\boldsymbol{u}), \\ 0, & i \notin \mathcal{T}_k(\boldsymbol{u}), \end{cases} \quad (7)$$

*where* $\mathcal{T}_k(\boldsymbol{u})$ *denotes the set of indices corresponding to the $k$ largest entries[2] of $\boldsymbol{u}$. Here $\lambda$, $\theta$ and $k$ are hyper-parameters subject to design choices.*

*They can be induced by the following choices of sparse regularizers:*

- *Case I:* $R(\boldsymbol{z}) = \|\boldsymbol{z}\|_1 + \iota_{\{\boldsymbol{z} \geq 0\}}(\boldsymbol{z})$, *then* $\text{prox}_{\lambda R} = \text{ReLU}_\lambda$;

- *Case II:* $R(\boldsymbol{z}) = \|\boldsymbol{z}\|_0 + \iota_{\{\boldsymbol{z} \geq 0\}}(\boldsymbol{z})$, *then* $\text{prox}_{\lambda R} = \text{JumpReLU}_{\sqrt{2\lambda}}$;

- *Case III:* $R(\boldsymbol{z}) = \iota_{\{\|\boldsymbol{z}\|_0 \leq k, \boldsymbol{z} \geq 0\}}(\boldsymbol{z})$, *then* $\text{prox}_{\lambda R}(\boldsymbol{u}) = \text{TopK}_k(\boldsymbol{u})$.

*Here $\iota_A$ is the indicator function of set $A$, i.e., $\iota_A(\boldsymbol{z}) = 0$ if $\boldsymbol{z} \in \mathbb{A}$ and $\iota_A(\boldsymbol{z}) = +\infty$ if $\boldsymbol{z} \notin \mathbb{A}$, and $\boldsymbol{z} \geq 0$ means $z_i \geq 0$ for all $i$.*

A detailed proof is provided in the Appendix C. Note that $\text{ReLU}_\lambda$ reduces to the standard ReLU when $\lambda \to 0$. The operators $\text{ReLU}_\lambda$ and $\text{JumpReLU}_\theta$ are commonly referred to as soft thresholding and hard thresholding (except restricted to the nonnegative orthant), respectively, in signal and image processing, where they are used to enforce sparsity (Foucart, 2011; Acuña et al., 2020). The TopK operator in (7) follows the original formulation in Gao et al. (2025), which includes an additional ReLU to ensure nonnegative activations. Nevertheless, if $\boldsymbol{u}$ has at least $k$ nonnegative entries—which is typically the case since $k$ is much smaller than the ambient dimension $s$—then the ReLU inside TopK is redundant, and the operator simply retains the largest $k$ entries while setting the rest to zero. This phenomenon is also observed in Gao et al. (2025), where the training curves were found to be indistinguishable. In a nutshell, Lemma 1 establishes that several prevalent non-linearities in SAEs, including ReLU, JumpReLU, and TopK, are precisely the proximal operators of sparse-enforcing regularizers.

**One-step proximal gradient method leads to Sparse Autoencoders.** With Lemma 1, applying a one-step proximal gradient method to the sparse coding problem naturally leads to SAEs. Specifically, (6) defines a mapping from an input representation $\boldsymbol{x}$ to a sparse code $\boldsymbol{z}$, which is then decoded to reconstruct the original representation, formally given by

$$\text{encoder: } \boldsymbol{z} = \text{prox}_{\lambda R}\left(\boldsymbol{W}^\top\boldsymbol{x} + \boldsymbol{b}_\text{e}\right), \quad \text{decoder: } \widehat{\boldsymbol{x}} = \boldsymbol{D}\boldsymbol{z} + \boldsymbol{b}. \quad (8)$$

---

[2]In case $k$ largest components are not uniquely defined, one can choose among them—for example, by selecting the components with the smallest indices—to ensure exactly $k$ entries are kept.

Choosing different regularizers $R$ as in Lemma 1 yields different variants of SAEs, including the vanilla version with ReLU (Cunningham et al., 2023), a version with JumpReLU (Rajamanoharan et al., 2025), and one with TopK (Gao et al., 2025). For simplicity, we refer to these as ReLU SAE, JumpReLU SAE, and TopK SAE, respectively. This observation situates diverse SAE architectures within a unified proximal framework, where each activation function is interpreted as the proximal map for a specific regularizer $R$. Consequently, design choices for SAEs correspond directly to the selection of an implicit sparsity-inducing penalty, which in turn provides a principled basis for comparing and extending these models. For instance, our analysis in Lemma 1 shows that ReLU SAE corresponds to the $\ell_1$ norm regularizer (a convex relaxation of sparsity) with weight $\lambda \to 0$, whereas JumpReLU and TopK correspond directly to the sparsity-inducing $\ell_0$ norm regularizers with a non-vanishing $\lambda$, thereby enforcing stronger sparsity. This provides a principled explanation for the improved performance of JumpReLU and TopK over ReLU observed in (Rajamanoharan et al., 2025; Gao et al., 2025).

Substituting (6) into (3) yields the training objective for SAEs (Cunningham et al., 2023; Rajamanoharan et al., 2025; Gao et al., 2025):

$$\min_{\substack{\boldsymbol{D}, \boldsymbol{W} \in \mathbb{R}^{d \times P} \\ \boldsymbol{b} \in \mathbb{R}^d, \boldsymbol{b}_{\mathrm{e}} \in \mathbb{R}^P}} \mathbb{E}_{\boldsymbol{x}} \left[ \frac{1}{2} \left\| \boldsymbol{x} - (\boldsymbol{D}\boldsymbol{z} + \boldsymbol{b}) \right\|_2^2 + \lambda R(\boldsymbol{z}), \text{ where } \boldsymbol{z} = \mathrm{prox}_{\lambda R}\left( \boldsymbol{W}^\top \boldsymbol{x} + \boldsymbol{b}_{\mathrm{e}} \right) \right]. \quad (9)$$

In practice, the two instances of $\lambda$ in (8) may be decoupled to provide additional flexibility for hyper-parameter tuning.

The use of a parameterized encoder is a key design choice that circumvents the challenging non-convex optimization in the original dictionary learning formulation (3), which requires simultaneous optimization over the sparse codes $\boldsymbol{z}$ and the dictionary parameters $\boldsymbol{D}$ and $\boldsymbol{b}$. By decoupling this joint optimization, SAEs yield a more tractable training procedure. The encoder arises as a single proximal gradient step, augmented with uncoupled, learnable parameters for the dictionary and bias to reduce the approximation error relative to exact sparse coding. Consequently, the training problem (8) can be efficiently solved via stochastic gradient descent, and SAEs can be implemented efficiently at inference time, making them attractive for interpretability research.

This perspective also provides a principled foundation for developing SAE variants with improved performance. For example, by incurring additional computational cost, one may extend (6) to multi-step variants, yielding multi-layer encoders (Tolooshams & Ba, 2022) that produce more accurate sparse codes and potentially capture finer-grained structure in the representation space. We leave this direction to future work. In the next subsection, we turn to SAEs induced by alternative sparsity regularizers.

## 2.3 Beyond non-negativity: Sparse Autoencoders with AbsTopK

The proximal perspective developed above suggests that design choices for SAEs can be interpreted as the selection of the sparsity-inducing penalty. While this view explains their sparsity-inducing effect, it also reveals a fundamental limitation of current SAEs in Equation (7): they prompt sparsity but also enforce non-negativity, discarding half of the representation space. As many semantic axes are naturally bidirectional (e.g., *male v.s. female*, *positive v.s. negative sentiment*), restricting sparse codes to be non-negative fragments these concepts into two separate directions or collapses one side entirely.

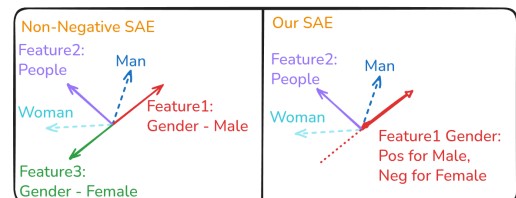

Figure 2: Toy example where *man* $\approx$ male + people and *woman* $\approx$ female + people: a non-negative SAE needs two separate gender features, whereas AbsTopK uses one signed gender feature.

**Fragmentation of Conventional SAE** To formalize this, consider a single semantic concept direction $\boldsymbol{h}$ in (2) represented by a dictionary atom $\boldsymbol{d} \in \mathbb{R}^d$. An ideal sparse code would represent concepts along this axis as $\alpha \boldsymbol{d}$, where the sign of the scalar $\alpha$ encodes directionality. However, under the non-negativity constraint $\boldsymbol{z} \geq 0$, this is impossible. Instead, a standard SAE must allocate two distinct dictionary atoms, $\boldsymbol{d}_i$ and $\boldsymbol{d}_j$,

oriented in opposite directions, with nonnegative activations $z_i \geq 0$ and $z_j \geq 0$ respectively. Each atom is activated only for one direction, leading to a fragmented representation that arises directly from the non-negativity constraint. This fragmentation is a direct consequence of the non-negativity constraint.

**Removing non-negativity as a remedy.**    To address this issue, we propose using a sparse regularizer without the non-negativity constraint. Different variants of sparse regularizers can be considered, with representative examples discussed in Lemma 1. In this work, we adopt the $\ell_0$ norm due to its simplicity and its direct connection to sparsity. Specifically, in the dictionary learning formulation (3), we use the regularizer $R(\boldsymbol{z}) = \iota_{\{\|\boldsymbol{z}\|_0 \leq k\}}$ which removes the non-negativity constraint present in the TopK-inducing regularizer. The corresponding proximal operator is

$$\mathrm{prox}_{\lambda R}(\boldsymbol{u}) = \arg\min_{\boldsymbol{z} \in \mathbb{R}^d} \tfrac{1}{2}\|\boldsymbol{u} - \boldsymbol{z}\|_2^2 \quad \text{s.t.} \quad \|\boldsymbol{z}\|_0 \leq k, \tag{10}$$

whose closed-form solution is further given by

$$\big(\mathrm{prox}_R(\boldsymbol{u})\big)_i = (\mathrm{AbsTopK}_k(\boldsymbol{u}))_i = \begin{cases} u_i, & i \in \mathcal{H}_k(\boldsymbol{u}), \\ 0, & i \notin \mathcal{H}_k(\boldsymbol{u}), \end{cases} \tag{11}$$

where $\mathcal{H}_k$ denotes the indices of the $k$ largest (in modulus) components[3]. In words, this operator preserves the $k$ largest-magnitude components of a vector and sets all others to zero. In the compressive sensing literature, it is referred to as the *hard thresholding operator* (Foucart, 2011). Here, we refer to it as Absolute TopK (AbsTopK) to distinguish it from the TopK operator commonly used in SAE.

This principle of hard thresholding can also be applied to JumpReLU, introducing a threshold on both positive and negative activations. This achieves a similar effect by eliminating small-magnitude features and enforcing sparsity. However, to isolate and directly test our core hypothesis, the value of representing concepts along a bipolar axis, this work focuses on AbsTopK, as it provides the most direct implementation of a global $k$-sparsity constraint. We remain JumpReLU variants for future investigation.

**AbsTopK SAE.**    Following the derivation in the previous section, we integrate the AbsTopK nonlinearity operator into the framework (8) to obtain a new SAE architecture, which we term AbsTopK SAE:

$$\boldsymbol{z} = \mathrm{AbsTopK}(\boldsymbol{W}^\top \boldsymbol{x} + \boldsymbol{b}_\mathrm{e}), \; \widehat{\boldsymbol{x}} = \boldsymbol{D}\boldsymbol{z} + \boldsymbol{b}. \tag{12}$$

The overall training problem becomes

$$\min_{\substack{\boldsymbol{D}, \boldsymbol{W} \in \mathbb{R}^{d \times P} \\ \boldsymbol{b} \in \mathbb{R}^d, \boldsymbol{b}_\mathrm{e} \in \mathbb{R}^P}} \mathbb{E}_{\boldsymbol{x}} \left[ \frac{1}{2}\big\|\boldsymbol{x} - (\boldsymbol{D}\boldsymbol{z} + \boldsymbol{b})\big\|_2^2, \text{ where } \boldsymbol{z} = \mathrm{AbsTopK}(\boldsymbol{W}^\top \boldsymbol{x} + \boldsymbol{b}_\mathrm{e}) \right]. \tag{13}$$

By design, AbsTopK preserves both positive and negative activations, enabling a single feature to capture contrastive concepts along a unified semantic axis. This simple modification circumvents the fragmentation induced by non-negativity constraints, and yields features that more faithfully reflect the bidirectional structure of semantic representations. Importantly, we do not claim that every feature should realize a perfectly symmetric semantic axis. When a concept is naturally bipolar, the model should be able to represent it with a single bidirectional feature, rather than having such features ruled out by construction. At the same time, the formulation fully supports unipolar concepts, these can simply make use of the positive side of the feature to sparsely encode the hidden states.

## 3    EXPERIMENTS: EMPIRICAL VALIDATION OF SAE BEHAVIOR

To empirically validate our theoretical claims and demonstrate the practical advantages of the AbsTopK operator, we perform a suite of experiments which involve training JumpReLU, TopK, and AbsTopK SAEs on `monology/pile-uncopyrighted` (Gao et al., 2020) across multiple models (Radford et al., 2019; Biderman et al., 2023; Team, 2024; Yang et al., 2025). To compare the

---

[3]Similarly, if the $k$ largest components are not uniquely defined, one can, for instance, select those with the smallest indices to ensure exactly $k$ entries are retained.

different SAEs, we evaluate their performance along several dimensions: (i) reconstruction quality on base datasets, (ii) effectiveness on a range of steering tasks, and (iii) impact on general capabilities of the models. For further experimental details and extended results, we refer the reader to Appendix B.

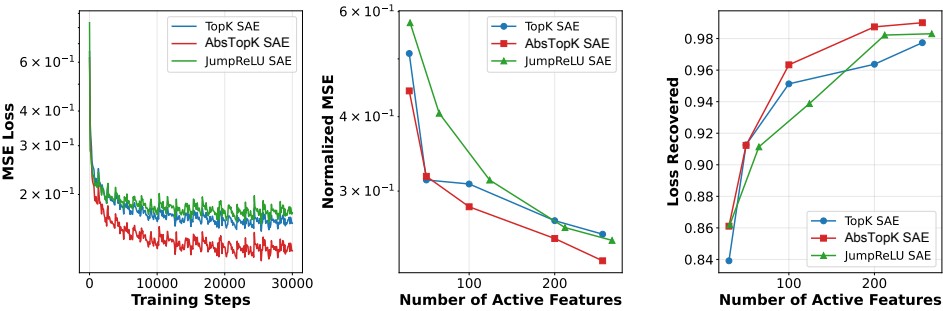

Figure 3: **Performance comparison of JumpReLU, TopK, and AbsTopK SAEs on Qwen3 4B Layer 20**, showing **(a)** MSE Training Loss, **(b)** Normalized MSE, and **(c)** Loss Recovered. Additional results across models and layers are provided in Appendix D.

## 3.1 UNSUPERVISED METRICS

This section presents a comparative evaluation of SAE architectures, utilizing a suite of complementary metrics engineered to assess distinct facets of model performance. The investigation encompasses three primary analyses: (a) an examination of the training mean squared error (MSE) to evaluate optimization stability and convergence rates; (b) the measurement of normalized reconstruction error as a function of feature sparsity to ascertain representational fidelity; and (c) a relative cross-entropy loss recovered score to determine the preservation of language modeling performance. For Topk and AbsTopK, sparsity is explicitly controlled by directly specifying the number of active features $k$; in contrast, for JumpReLU, sparsity is varied by manually adjusting the threshold parameter $\theta$, thereby simulating different sparsity levels.

The normalized reconstruction error in (b) is defined as $\text{nMSE}(\boldsymbol{x}, \hat{\boldsymbol{x}}) = \|\boldsymbol{x} - \hat{\boldsymbol{x}}\|_2^2 / \|\boldsymbol{x}\|_2^2$ (Gao et al., 2025), thereby controlling for scale differences across representations. The Loss Recovered score in (c) measures how well SAE reconstructions preserve predictive performance (Karvonen et al., 2025), defined as $(H^* - H_0)/(H_{\text{orig}} - H_0)$, where $H_{\text{orig}}$ is the cross-entropy of the original model, $H^*$ that after substitution, and $H_0$ under zero-ablation, with values closer to one indicating better preservation.

AbsTopK achieves the most favorable behavior and consistently attains lower reconstruction error across most sparsity levels while inducing only minor cross-entropy degradation. This advantage is explained by the expressiveness of the underlying constraints. TopK and JumpReLU enforce non-negativity, inducing a conical decomposition that tends to split inherently bidirectional concepts across multiple features. AbsTopK instead allows signed activations, so a single feature can encode opposite concepts via its sign, yielding a more compact and interpretable linear decomposition of the latent space. As we show in subsequent qualitative analyses, this bidirectional capacity leads to dictionary atoms that more closely align with the conceptual structure of the model's representations.

## 3.2 RESULTS ON PROBE AND STEERING TASKS

To assess the utility of learned SAE features for model control, a comprehensive benchmarking evaluation was conducted across a diverse suite of steering and probing tasks. These tasks were specifically designed to probe various dimensions of feature quality, from basic concept representation to the capacity for precise interventional control. A detailed methodological overview for each metric is provided in the Appendix E.

The empirical results, as shown in Figure 4, demonstrate the superiority of the AbsTopK methodology. Across the entire suite of evaluated tasks, AbsTopK SAE outperforms both the TopK SAE

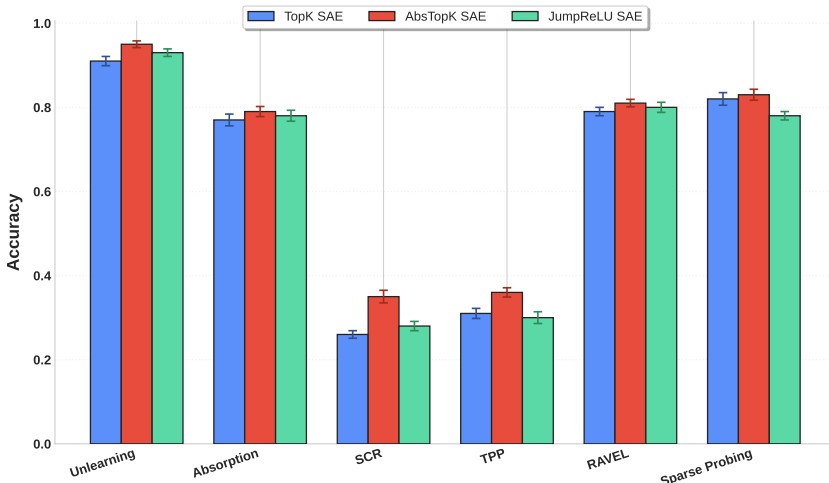

Figure 4: **Performance comparison of SAE variants (TopK, AbsTopK, and JumpReLU) across tasks on Qwen3-4B Layer 18.** For all tasks, higher scores indicate better performance; the Unlearning and Absorption scores have been transformed as 1−original score to maintain this consistency. We report the mean across five runs (random seeds 40–44), with error bars indicating the standard deviation. For more details, see Appendix E.

and JumpReLU SAE baselines. This performance advantage is especially conspicuous in bidirectional steering metrics, such as SCR, which directly quantify the reliability of interventions. In these critical evaluations, AbsTopK shows marked improvements over the alternatives.

We posit that this consistent outperformance is directly attributable to the core mechanism of the AbsTopK methodology: the retention of both positive and negative feature activations. Unlike TopK approaches, which enforce a hard sparsity constraint that discards all but the most prominent positive activations, AbsTopK preserves a richer, more complete semantic representation. This retention is critical for interventions that require nuanced and bidirectional control. By encoding not only the presence of a concept but also its negation or semantic opposition, AbsTopK features provide a more robust and granular basis for manipulation.

## 3.3 EMPIRICAL RESULTS ON STEERING VS. UTILITY

Table 1: **Performance comparison on MMLU (↑) and HarmBench (↑) across steering methods.** Entries show the absolute score; colored values in parentheses indicate the change relative to the unsteered **Original** model (red: improvement, blue: drop). The best result among all methods for each metric is highlighted in **bold**.

| Model | Layer | Metric | Original | ReLU SAE | JumpReLU SAE | TopK SAE | AbsTopK SAE | DiM |
|---|---|---|---|---|---|---|---|---|
| Qwen3 4B | 18 | MMLU | 77.3 | (-2.9) 74.4 | (-2.3) 75.0 | (-2.1) 75.2 | (-1.4) **75.9** | (-1.5) 75.8 |
| | | HarmBench | 17.0 | (+61.5) 78.5 | (+62.1) 79.1 | (+61.2) 78.2 | (+64.3) **81.3** | (+63.6) 80.6 |
| | 20 | MMLU | 77.3 | (-1.5) 75.8 | (-1.6) 75.7 | (-2.3) 75.0 | (-0.9) **76.4** | (-0.9) **76.4** |
| | | HarmBench | 17.0 | (+60.2) 77.2 | (+61.5) 78.5 | (+60.0) 77.0 | (+62.0) 79.0 | (+63.0) **80.0** |
| Gemma2 2B | 12 | MMLU | 52.2 | (-2.9) 49.3 | (-3.4) 48.8 | (-3.1) 49.1 | (-0.9) **51.3** | (-1.2) 51.0 |
| | | HarmBench | 19.0 | (+48.9) 67.9 | (+50.5) 69.5 | (+50.8) 69.8 | (+51.2) 70.2 | (+51.8) **70.8** |
| | 16 | MMLU | 52.2 | (-2.2) 50.0 | (-4.0) 48.2 | (-3.7) 48.5 | (-1.2) **51.0** | (-1.4) 50.8 |
| | | HarmBench | 19.0 | (+50.9) 69.9 | (+50.8) 69.8 | (+51.2) 70.2 | (+52.7) 71.7 | (+53.0) **72.0** |
| Llama3.1 8B | 24 | MMLU | 66.7 | (-2.5) 64.2 | (-2.5) 64.2 | (-1.7) 65.0 | (-0.9) **65.8** | (-1.3) 65.4 |
| | | HarmBench | 15.2 | (+75.0) 90.2 | (+74.7) 89.9 | (+74.0) 89.2 | (+76.1) 91.3 | (+77.2) **92.4** |
| Gemma3 12B | 6 | MMLU | 74.5 | (-2.7) 71.8 | (-2.0) 72.5 | (-1.5) 73.0 | (-1.3) **73.2** | (-1.4) 73.1 |
| | | HarmBench | 16.6 | (+47.8) 64.4 | (+45.5) 62.1 | (+46.2) 62.8 | (+49.0) **65.6** | (+48.8) 65.4 |
| | 40 | MMLU | 74.5 | (-3.2) 71.3 | (-2.5) 72.0 | (-3.5) 71.0 | (-1.8) **72.7** | (-2.0) 72.5 |
| | | HarmBench | 16.6 | (+70.2) 86.8 | (+72.0) 88.6 | (+70.7) 87.3 | (+72.6) 89.2 | (+73.4) **90.0** |

To more comprehensively characterize the safety–utility trade-off, we evaluate steering across four model and intervene at multiple layers spanning early, middle, and late blocks. This diversity in both architectures and intervention depths allows us to test whether our conclusions are robust to model scale and to the choice of steering layer, rather than being an artifact of a single model configuration.

Model steering confronts a fundamental tradeoff: enhancing specific behaviors often degrades general capabilities. It has often been assumed in prior literature that DiM interventions are more effective for specific concept manipulation than SAEs despite their reliance on labeled data and limitation to extracting only a single concept vector (Arditi et al., 2024; Wu et al., 2025; Zhu et al., 2025). To systematically evaluate this trade-off, we conducted an empirical study measuring general capability preservation via the MMLU benchmark (Hendrycks et al., 2021) and safety alignment using HarmBench (Mazeika et al., 2024). For this evaluation, we focus on Qwen and Gemma models, as smaller models, Pythia-70M and GPT-2 Small, only have very low score on MMLU benchmark.

As shown in Table 1, the empirical results indicate that conventional SAE steering methods successfully improve safety metrics but at a detriment to general performance. In contrast, the proposed AbsTopK methodology achieves a more optimal balance between these competing objectives. It facilitates substantial enhancements in safety alignment on HarmBench while simultaneously mitigating the degradation of MMLU scores. Compared to DiM, AbsTopK is competitive on safety, sometimes slightly lower, but consistently retains more general ability. This pattern highlights that carefully designed SAE steering can rival and, in some cases, surpass intervention strategies that rely on labeled data.

### 3.4 BIDIRECTIONAL SEMANTIC AXES IN ABSTOPK VS. TOPK

Table 2: LLM-based automatic interpretation of AbsTopK and TopK features on Gemma-2-2B. For each layer and method, we report the proportion of features in three semantic categories: double-sided, single-sided, and no clear meaning. The row marked with ↪ Opposite meaning gives the subset of double-sided features whose two polarities express opposite semantics.

| Category | Layer 12 (%) | | Layer 16 (%) | |
|---|---|---|---|---|
| | AbsTopK | TopK | AbsTopK | TopK |
| Double-sided meaning (all) | 29.7 | 5.3 | 31.2 | 4.1 |
| ↪ Opposite meaning | 20.2 | 2.6 | 21.5 | 1.8 |
| Single-sided meaning | 56.4 | 78.8 | 57.8 | 80.3 |
| No clear meaning | 13.9 | 15.9 | 11.0 | 15.6 |

To quantify the bidirectionality, we apply Gemini 2.5 Flash to classify each feature into three categories: double-sided meaning, where both polarities are judged meaningful; single-sided meaning, where only one polarity is meaningful; and no clear meaning. Within the double-sided group, we further identify a subset of opposite meaning features whose positive and negative activations are judged to express opposing semantics.

Table 2 summarizes the distribution of these categories. Across both layers, the fraction with no clear meaning is comparable for AbsTopK and TopK, indicating that the bidirectional features in AbsTopK does not arise from noisier features. The nonzero mass of opposite-meaning features under TopK suggests that the underlying representation already supports bidirectional semantic axes, but only weakly exploits them. By relaxing the non-negativity constraint, AbsTopK converts part of this single-sided inventory into semantic directions with two meaningful ends.

## 4 CONCLUSION

This work identifies the non-negativity constraint in SAEs as a core cause of semantic feature fragmentation. In response, we introduce the AbsTopK operator, which replaces this constraint with direct k-sparsity enforced via an $\ell_0$ proximal operator. This modification enables single features to capture bipolar semantics, and our empirical results confirm that AbsTopK yields reconstructions of superior compactness and fidelity. Our work pioneers a shift towards bipolar sparse representations and suggests future research into more efficient, neurally-plausible approximations of the $\ell_0$ operator for large-scale models.

### ACKNOWLEDGEMENTS

We acknowledge support from the National Science Foundation under grants IIS-2312840, IIS-2402952, IIS-2301599, CMMI-2301601, and DMS-2529302. We thank members of the OSU CSE

community for valuable discussions and feedback. We thank the Ohio Supercomputer Center for providing the computational resources.

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

## A    RELATED WORKS

**Sparse Dictionary Learning**    The dictionary learning problem (3) is highly nonconvex. Over the past decades, numerous heuristic methods have been proposed to solve it efficiently (Cai et al., 2016; Aharon et al., 2006; Tošić & Frossard, 2011). Significant effort has also been devoted to addressing the nonconvexity and establishing theoretical guarantees for exact recovery, including approaches based on convex relaxation (Spielman et al., 2012), semidefinite programming (Barak et al., 2015), landscape analysis of simplified formulations that estimate one atom at a time (Sun et al., 2016; Gilboa et al., 2019; Zhu et al., 2019; Qu et al., 2020), alternating minimization (Arora et al., 2015; Chatterji & Bartlett, 2017), as well as global convergence guarantees to critical points (Bao et al., 2014; Hastie et al., 2015; Bao et al., 2016). Subsequent work has explored unrolled methods (Malézieux et al., 2022; Tolooshams & Ba, 2022; Chen et al., 2022) and over-parameterization (Sulam et al., 2022), analyzed convergence for solving the sparse coding problem under the assumption of a fixed dictionary (Gregor & LeCun, 2010; Tang et al., 2020; Massoli et al., 2024), and examined gradient stability (Tolooshams & Ba, 2022; Malézieux et al., 2022).

Building on these works, we re-examine recently popularized SAEs for interpretability through the lens of unrolled dictionary learning. This perspective reveals a direct correspondence between activation functions and the proximal mappings of sparse regularizers, thereby situating SAEs within the broader framework of dictionary learning. Leveraging this connection, we introduce a new activation, *AbsTopK*, which removes the non-negativity constraint imposed in prior designs and enables a single dictionary feature to encode bidirectional semantic axes. Our principal contribution is thus a formal alignment of SAE architecture with the intrinsic geometry of semantic representation—distinct from classical theories focused on signal recovery.

**Mechanistic interpretability**

SAEs have become a central tool in mechanistic interpretability, serving as a dictionary learning approach for concept-level explanations (Kim et al., 2018). Several architectures have been proposed, including ReLU, TopK, JumpReLU, gated, Batch TopK, and ProLU SAEs (Bricken et al., 2023; Gao et al., 2025; Rajamanoharan et al., 2025; 2024b; Bussmann et al., 2024; O'Neill et al., 2025), and have been shown to capture a wide range of interpretable features, from refusal, gender, and writing script (Bricken et al., 2023; Templeton et al., 2024; Hegde, 2024) to visual structure and protein representations (Thasarathan et al., 2025; Simon & Zou, 2024).

At the same time, recent work has highlighted important limitations of the SAE paradigm. Prompting-based interventions can outperform SAE-based control (Wu et al., 2025; Bhalla et al., 2025; Menon et al., 2024); other studies question the assumption that concepts are well captured by single linear features, showing that representations can be multidimensional or nonlinear (Engels et al., 2024; 2025; Peng et al., 2025; Wang et al., 2025). Moreover, SAEs can be algorithmically unstable: models trained on the same data with different random seeds may yield divergent dictionaries and inconsistent interpretations (Ayonrinde et al., 2024; Kissane et al., 2024; Colin et al., 2025). These observations suggest that, while SAEs are promising for interpretability, their current formulations are fragile and lack a canonical notion of representation.

Motivated by these challenges, recent theoretical work (Chen et al., 2025; Cui et al., 2025) investigates when standard non-negative SAEs can provably recover ground-truth features under sparse, non-negative latent codes, providing a principled justification for non-negativity in unipolar settings. Our framework is complementary and tailored to the mixed-sign, bidirectional structure observed in LLM representations: from a proximal-gradient viewpoint, we interpret SAE nonlinearities as proximal operators with an implicit non-negativity constraint, and relax this constraint to obtain AbsTopK, which preserves the same sparse coding objective while enabling bidirectional semantic axes and reducing to standard SAEs in the non-negative limit.

## B    EXPERIMENTAL SETUP

In this appendix, we describe the architecture and training setup of our SAEs. For all experiments, we trained on the `monology/pile-uncopyrighted` (Gao et al., 2020) dataset.

Architecturally, the SAEs are comprised of a single, overcomplete hidden layer which incorporates a sparsifying nonlinearity. The encoder component projects residual activations into a latent space

of higher dimensionality, while the decoder component reconstructs the original residual dimension from these latent representations. A fixed expansion factor of 16 was uniformly applied across all models.

For comparative analysis, three distinct variants of the SAE were trained: TopK, AbsTopK, and JumpReLU. In the TopK and AbsTopK configurations, exact k-sparsity was enforced upon the latent representation, with the sparsity hyperparameter, k, and the specific layers targeted for intervention being systematically selected for each foundational model:

- `EleutherAI/pythia-70m` (Biderman et al., 2023): $k = 51$, layers: 3, 4.
- `google/gemma-2-2b` (Team, 2024): $k = 230$, layers: 12, 16.
- `Qwen/Qwen3-4B` (Yang et al., 2025): $k = 256$, layers: 18, 20.
- `openai-community/gpt2` (Radford et al., 2019): $k = 76$, layers: 6, 8.

Here, $k$ was set to approximately one-tenth of the hidden dimension for each model, and the intervention layers were selected from the middle of the network to capture representative latent features (Arditi et al., 2024). In contrast, the JumpReLU models adopted the same configuration as in prior work (Rajamanoharan et al., 2025; Bussmann et al., 2024).

The optimization for all models was performed using the Adam algorithm over a duration of 30,000 training steps, with a consistent batch size of 4096. A learning rate of 3e-4 was configured, complemented by Adam's momentum parameters, $\beta_1 = 0.9$, and $\beta_2 = 0.99$. And we used a bandwidth parameter of 0.001 across all experiments.

## C  PROOF OF LEMMA 1

*Proof.* We prove the result by deriving the proximal operator corresponding to each regularizer separately.

**Case I: ReLU.**  Note that $R(\boldsymbol{z})$ is separable as

$$R(\boldsymbol{z}) = \|\boldsymbol{z}\|_1 + \iota_{\{\boldsymbol{z} \geq 0\}}(\boldsymbol{z}) = \sum_i \left( |z_i| + \iota_{\{z_i \geq 0\}}(z_i) \right),$$

which implies that the proximal operator is also separable, i.e., $(\text{prox}_{\lambda R}(\boldsymbol{u}))_i$ is equivalent to the following scalar proximal problem

$$
\begin{aligned}
\text{prox}_{\lambda R}(u) &= \arg\min_{z \in \mathbb{R}} \ \frac{1}{2}(z - u)^2 + \lambda|z| + \iota_{\{z \geq 0\}}(z) \\
&= \arg\min_{z \geq 0} \ \frac{1}{2}(z - u)^2 + \lambda z \\
&= \max\{u - \lambda, 0\}.
\end{aligned}
$$

Therefore, the proximal operator induces the ReLU operator, with a shift by $\lambda$:

$$\boxed{(\text{prox}_{\lambda R}(\boldsymbol{u})) \ = \ \max\{\boldsymbol{u} - \lambda, 0\},}$$

which reduces to the standard ReLU when $\lambda \to 0$. In this case, however, the operator no longer encourages sparsity. When $\lambda > 0$, the effect is equivalent to introducing a bias term that suppresses small activations and thereby promotes sparsity. In practice, this restriction can be relaxed: during training, gradient descent can learn a separate bias parameter for each entry.

**Case II: JumpReLU.**  Similarly, $R(\boldsymbol{z})$ is also separable as

$$R(\boldsymbol{z}) = \|\boldsymbol{z}\|_0 + \iota_{\boldsymbol{z} \geq 0}(\boldsymbol{z}) = \sum_i \left( \mathbf{1}(z_i \neq 0) + \iota_{\{z_i \geq 0\}}(z_i) \right).$$

where $\mathbf{1}(z_i \neq 0) = \begin{cases} 1, z_i \neq 0, \\ 0, z_i = 0. \end{cases}$ Thus, it suffices to first consider the following scalar proximal operator

$$\text{prox}_{\lambda R}(u) = \arg\min_{z \in \mathbb{R}} \; \frac{1}{2}(z-u)^2 + \lambda\mathbf{1}(z \neq 0) + \iota_{\{z \geq 0\}}(z)$$

$$= \arg\min_{z \geq 0} \; \underbrace{\frac{1}{2}(z-u)^2 + \lambda\mathbf{1}(z \neq 0)}_{\xi(z)}.$$

Note that within the region $z \geq 0$, $\xi$ achieve its minimum at either 0 or $u$. Setting $\xi(u) = \lambda = \xi(0) = \frac{1}{2}u^2$ yields $u = \sqrt{2\lambda}$. One can verify that $\xi$ achieves its minimum at $u$ when $u \geq \sqrt{2\lambda}$, and at 0 otherwise. Hence, the proximal operator induces the JumpReLU with parameter $\sqrt{2\lambda}$:

$$(\text{prox}_{\lambda R}(\boldsymbol{u}))_i = \begin{cases} u, & u \geq \sqrt{2\lambda}, \\ 0, & u < \sqrt{2\lambda}. \end{cases}$$

**Case III: TopK.** For this case, the corresponding proximal operator reduces to a Euclidean projection onto the feasible set:

$$\text{prox}_{\lambda R}(\boldsymbol{u}) = \arg\min_{\boldsymbol{z} \in \mathbb{R}^d} \; \frac{1}{2}\|\boldsymbol{u} - \boldsymbol{z}\|_2^2 \quad \text{s.t.} \quad \|\boldsymbol{z}\|_0 \leq k, \; \boldsymbol{z} \geq 0. \tag{14}$$

Given the quadratic objective and the non-negativity constraint, the optimal choice on any candidate support $S$ with $|S| \leq k$ is

$$z_i = \begin{cases} \max\{u_i, 0\}, & i \in S, \\ 0, & i \notin S. \end{cases} \tag{15}$$

Thus, the minimization problem reduces to selecting the index set $S$ that captures the $k$ largest nonnegative entries of $\boldsymbol{u}$. Formally, letting $\mathcal{T}_k(\boldsymbol{z})$ denote the set of indices corresponding to the $k$ largest entries of $\boldsymbol{z}$, the proximal operator becomes

$$[\text{prox}_{\lambda R}(\boldsymbol{u})]_i = \begin{cases} \max\{u_i, 0\}, & i \in \mathcal{T}_k(\boldsymbol{z}), \\ 0, & i \notin \mathcal{T}_k(\boldsymbol{z}). \end{cases} \tag{16}$$

$\square$

# D  UNSUPERVISED METRICS ON ALL MODELS

This section presents the unsupervised metrics from our model evaluations. We tested each model with a specific set of k values. For the Pythia model, we used k-values of 10, 20, 30, 40, and 50. The evaluation of the Gemma model involved k values of 30, 50, 100, 200, and 230. For the GPT model, the k values were 10, 30, 50, 60, and 76. Lastly, the Qwen model was tested with k values of 30, 50, 100, 200, and 256.

As shown in Figure 5, across the majority of evaluated models, we observe that AbsTopK achieves lower training MSE, reduced normalized reconstruction error, and better preservation of language modeling performance relative to both TopK and JumpReLU. This consistent advantage across these metrics provides evidence for the effectiveness and robustness of the AbsTopK method.. In particular, while TopK and JumpReLU sometimes exhibit competitive performance in isolated settings, AbsTopK maintains robustness across architectures and layers, thereby demonstrating the superiority of our proposed formulation.

# E  STEERING AND PROBE TASK ON ALL MODELS

## E.1  TASK DESCRIPTION

We provide an overview of the tasks employed in the SAEBench evaluation for SAEs. For detailed methodology, we refer readers to the original SAEBench paper (Karvonen et al., 2025).

Table 3: **Performance comparison of SAE variants across tasks on all other models and layers.** For all tasks, higher scores indicate better performance; the Unlearning and Absorption scores have been transformed as $1-$original score to maintain this consistency.

| Model | Method | Unlearning | Absorption | SCR | TPP | RAVEL | Sparse Probing |
|---|---|---|---|---|---|---|---|
| Gemma2-2B L12 | AbsTopK | 0.93 | 0.73 | 0.27 | 0.34 | 0.73 | 0.76 |
| | TopK | 0.88 | 0.76 | 0.20 | 0.29 | 0.70 | 0.71 |
| | JumpReLU | 0.90 | 0.75 | 0.22 | 0.30 | 0.71 | 0.73 |
| Gemma2-2B L16 | AbsTopK | 0.91 | 0.70 | 0.27 | 0.42 | 0.71 | 0.70 |
| | TopK | 0.89 | 0.68 | 0.21 | 0.36 | 0.74 | 0.67 |
| | JumpReLU | 0.94 | 0.69 | 0.23 | 0.39 | 0.72 | 0.69 |
| Pythia-70M L3 | AbsTopK | 0.75 | 0.54 | 0.20 | 0.22 | 0.64 | 0.66 |
| | TopK | 0.71 | 0.47 | 0.15 | 0.14 | 0.62 | 0.60 |
| | JumpReLU | 0.73 | 0.50 | 0.17 | 0.21 | 0.63 | 0.61 |
| Pythia-70M L4 | AbsTopK | 0.79 | 0.53 | 0.21 | 0.23 | 0.68 | 0.57 |
| | TopK | 0.72 | 0.50 | 0.16 | 0.21 | 0.69 | 0.61 |
| | JumpReLU | 0.77 | 0.51 | 0.17 | 0.20 | 0.61 | 0.62 |
| GPT2-small L6 | AbsTopK | 0.74 | 0.66 | 0.18 | 0.22 | 0.60 | 0.54 |
| | TopK | 0.80 | 0.63 | 0.14 | 0.19 | 0.57 | 0.50 |
| | JumpReLU | 0.77 | 0.65 | 0.15 | 0.20 | 0.58 | 0.52 |
| GPT2-small L8 | AbsTopK | 0.75 | 0.67 | 0.23 | 0.28 | 0.51 | 0.59 |
| | TopK | 0.71 | 0.67 | 0.15 | 0.20 | 0.48 | 0.55 |
| | JumpReLU | 0.73 | 0.67 | 0.18 | 0.23 | 0.49 | 0.57 |
| Qwen3-4B L18 | AbsTopK | 0.95 | 0.79 | 0.35 | 0.36 | 0.81 | 0.83 |
| | TopK | 0.91 | 0.77 | 0.26 | 0.31 | 0.79 | 0.82 |
| | JumpReLU | 0.93 | 0.78 | 0.28 | 0.30 | 0.80 | 0.78 |
| Qwen3-4B L20 | AbsTopK | 0.95 | 0.80 | 0.32 | 0.45 | 0.85 | 0.81 |
| | TopK | 0.92 | 0.77 | 0.27 | 0.36 | 0.76 | 0.84 |
| | JumpReLU | 0.93 | 0.78 | 0.29 | 0.39 | 0.81 | 0.83 |

### E.1.1 FEATURE ABSORPTION

Sparsity incentives can cause a SAE to engage in feature absorption, a phenomenon where correlated features are merged into a single latent representation. This process arises when a direct implication exists between two concepts, such that concept $A$ always implies concept $B$. To reduce the number of active latents, the SAE might absorb the feature for $A$ into the latent for $B$. For example, a feature for "starts with S" could be absorbed into a more general latent for "short." While this merging improves computational efficiency, it compromises interpretability by creating gerrymandered features that represent multiple, distinct concepts.

To quantify feature absorption, we employ a first-letter classification task, following the methodology of previous studies (Chanin et al., 2025). First, a supervised logistic regression probe is trained on tokens containing only English letters to establish ground-truth feature directions. Next, K-sparse probing is applied to the SAE's latents to identify the primary latent corresponding to each feature, using a threshold of $\tau_{fs} = 0.03$ to account for potential feature splits. For test set tokens where main latents fail but the probe succeeds, additional SAE latents are included if they satisfy cosine similarity with the probe of at least $\tau_{ps} = 0.025$ and a projection fraction of at least $\tau_{pa} = 0.4$. All parameter values are chosen following the original SAEBench settings (Karvonen et al., 2025). To make the results more interpretable and such that higher values indicate stronger unlearning, we present the final scores as $1 -$ original value.

### E.1.2 UNLEARNING

SAEs are evaluated on their ability to selectively remove knowledge while maintaining performance on unrelated tasks (Farrell et al., 2025). We use the WMDP-bio dataset (Li et al., 2024) for unlearning and MMLU (Hendrycks et al., 2021) to assess general abilities.

The intervention methodology involves clamping selected WMDP-bio SAE feature activations to negative values whenever the corresponding features activate during inference. To evaluate broader model effects, we also measure performance on the MMLU benchmark (Hendrycks et al., 2021). The final evaluation reports the highest unlearning effectiveness on WMDP-bio while ensuring MMLU accuracy remains above 0.99, thereby quantifying optimal unlearning performance under constrained side effects. To make the results more interpretable and such that higher values indicate stronger unlearning, we present the final scores as $1 -$ original value.

### E.1.3 SPURIOUS CORRELATION REMOVAL (SCR)

SCR (Karvonen et al., 2024) evaluates the ability of SAEs to disentangle latents corresponding to distinct concepts. We conduct experiments on datasets known for spurious correlations, such as Bias in Bios (De-Arteaga et al., 2019) and Amazon Reviews (Hou et al., 2024), which contain two binary gender labels. For each dataset, we create a balanced set containing all combinations of profession (professor/nurse) and gender (male/female), as well as a biased set including only male+professor and female+nurse combinations. A biased classifier $C$ is first trained on the biased set and then debiased by ablating selected SAE latents.

We quantify SCR using the normalized evaluation score:

$$S_{\text{SHIFT}} = \frac{A_{\text{abl}} - A_{\text{base}}}{A_{\text{oracle}} - A_{\text{base}}}, \tag{17}$$

where $A_{\text{abl}}$ is the probe accuracy after SAE feature ablation, $A_{\text{base}}$ is the baseline accuracy before ablation, and $A_{\text{oracle}}$ is the skyline accuracy obtained by a probe trained directly on the desired concept. Higher $S_{\text{SHIFT}}$ values indicate more effective removal of spurious correlations. This score represents the proportion of improvement achieved through ablation relative to the maximum possible improvement, enabling fair comparison across classes and models.

### E.1.4 TARGETED PROBE PERTURBATION (TPP)

TPP (Marks et al., 2025) extends the SHIFT methodology to multiclass natural language processing datasets. For each class $c_i$ in a dataset, we select the most relevant SAE latents $L_i$. We then evaluate the causal effect of ablating $L_i$ on linear probes $C_j$ trained to classify each class $c_j$.

Let $A_j$ denote the accuracy of probe $C_j$ before ablation, and $A_{j \setminus i}$ the accuracy after ablating $L_i$. We define the accuracy change as

$$\Delta A_{j \setminus i} = A_{j \setminus i} - A_j. \tag{18}$$

The TPP score is then

$$S_{\text{TPP}} = \mathbb{E}_{i=j} \big[ \Delta A_{j \setminus i} \big] - \mathbb{E}_{i \neq j} \big[ A_{j \setminus i} \big], \tag{19}$$

which measures the extent to which ablating latents for class $i$ selectively degrades the corresponding probe while leaving other probes unaffected. A high TPP score thus indicates effective disentanglement of SAE latents.

### E.1.5 RAVEL

RAVEL (Chaudhary & Geiger, 2024) evaluates the ability of SAEs to disentangle features by testing whether individual latents correspond to distinct factual attributes. The dataset spans five entity types (cities, Nobel laureates, verbs, physical objects, and occupations), each with 400–800 instances and 4–6 attributes (e.g., cities have country, continent, and language), probed with 30–90 natural language and JSON prompt templates.

Evaluation proceeds in three stages: (i) filtering entity and attribute pairs that the model predicts reliably, (ii) identifying attribute and specific features using probes trained on latent representations, and (iii) computing a disentanglement score that averages *cause* and *isolation* metrics. The *cause* score measures whether intervening on a feature for attribute $A$ (e.g., setting Paris's country to Japan) correctly changes the prediction of $A$, while the *isolation* score verifies that other attributes $B$ (e.g., language = French) remain unaffected. A higher final score indicates stronger disentanglement of features.

### E.1.6 PROBING EVALUATION

We assess whether SAEs capture interpretable features through targeted probing tasks across five domains: profession classification, sentiment and product categorization , language identification, programming language classification, and topic categorization. Each dataset is partitioned into multiple binary classification tasks, yielding a total of 35 evaluation tasks.

For each task, we encode inputs with the SAE, apply mean pooling over non-padding tokens, and select the topk latents via maximum mean difference. A logistic regression probe is then trained on these representations and evaluated on held-out test data. To ensure comparability across tasks, we sample 4,000 training and 1,000 test examples per task, truncate inputs to 128 tokens, and, for GitHub, exclude the first 150 characters following Gurnee et al. (2023). We also compare mean and max pooling, finding mean pooling slightly superior. Datasets with more than two classes are subsampled into balanced binary subsets while maintaining a positive class ratio of at least 0.2.

### E.2 TASK PERFORMANCE

As shown in Table 3, we find that the AbsTopK methodology exhibits a superior level of performance relative to the comparative TopK and JumpReLU techniques across the evaluated models and layers.

In particular, the AbsTopK operator performs best on the majority of the evaluation metrics. While its performance is more competitive in a few areas, its dominant strength in the other key areas makes it a robust and highly effective sparsity operator according to these results. The method's strength appears to be model-agnostic, showcasing its general applicability.

## F STEERING METHODS FOR DIM AND SAES

In this section, we present methods for controlling specific concepts in model representations. For DiM, we introduce two intervention strategies: *activation addition*, to amplify a concept's effect, and *directional ablation*, to remove it from intermediate activations. For the HarmBench experiments, we specifically employ the activation addition method. Following this, we describe how similar steering can be achieved in SAEs through latent feature manipulation and ablation.

**Activation addition.** Given a concept vector $\boldsymbol{d}^{(l)}$ extracted from layer $l$, we can modulate the corresponding feature via a simple linear intervention. Concretely, for a specific input, we add the vector to the layer activations with the strength $\alpha$ to shift them toward the concept activation, thereby inducing the given concept:

$$\boldsymbol{x}^{(l)\prime} \leftarrow \alpha\boldsymbol{d}^{(l)} + \boldsymbol{x}^{(l)}. \tag{20}$$

This intervention is applied only at layer $l$ and across all token positions.

**Directional ablation.** To study the role of a particular direction $\boldsymbol{d}$ in the model's computation, we can remove it from the representations using directional ablation. Specifically, we zero out the component along $\boldsymbol{d}$ for every residual stream activation $\boldsymbol{x}$:

$$\boldsymbol{x}^{(l)\prime} \leftarrow \boldsymbol{x}^{(l)} - \alpha\boldsymbol{d}\boldsymbol{d}^\top\boldsymbol{x}^{(l)}. \tag{21}$$

This operation is applied to every activation $\boldsymbol{x}^{(l)}$, across all layers $l$, effectively preventing the model from encoding this direction in its residual stream.

**SAE Latent feature clamping.** For a target latent feature $z_i$ in the SAE feature vector $\boldsymbol{z}$, we can modulate its influence on model behavior by clamping it to a constant $c \in \mathbb{R}$. Denote a feature vector $\boldsymbol{z}$, and let $\boldsymbol{z}_{i,c}$ be the modified vector with $z_i$ replaced by $c$.

Define the clamping function $C_{i,c}$ as

$$[C_{i,c}(\boldsymbol{z})]_k = \begin{cases} z_k & \text{if } k \neq i, \\ c & \text{if } k = i, \end{cases} \qquad (22)$$

so that $C_{i,c}(\boldsymbol{z}) = \boldsymbol{z}_{i,c}$.

In conventional SAEs, this clamping strategy can be interpreted as a directional control: setting $c$ to a negative value suppresses the corresponding concept, while a positive $c$ encourages it. We adopt a similar approach to perform steering in our framework, using clamping to directly modulate individual latent features and thereby control the presence or absence of specific semantic concepts in the reconstructed representation.

## G  AUTOMATIC INTERPRETABILITY METRICS

Table 4: Automated Interpretability accuracy and PS-EVAL F1 for AbsTopK and TopK SAEs on meta-llama/Llama-3.1-8B across layers and activation types.

| Activation type | SAE | Automated Interpretability | | PS-EVAL F1 | |
|---|---|---|---|---|---|
| | | Layer 6 | Layer 28 | Layer 6 | Layer 28 |
| Attention out | AbsTopK | **0.82** | **0.81** | **0.74** | **0.62** |
| | TopK | 0.79 | 0.77 | 0.69 | 0.61 |
| MLP out | AbsTopK | **0.84** | **0.83** | **0.68** | **0.47** |
| | TopK | 0.83 | **0.83** | 0.65 | 0.44 |
| Residual stream | AbsTopK | **0.86** | **0.87** | **0.81** | **0.58** |
| | TopK | 0.81 | 0.80 | 0.76 | 0.56 |
| Transcoder | AbsTopK | **0.79** | **0.80** | **0.74** | **0.61** |
| | TopK | **0.79** | 0.78 | 0.72 | 0.57 |

We also evaluate interpretability directly using automatic metrics on meta-llama/Llama-3.1-8B, applying Automated Interpretability and PS-EVAL to feature dictionaries learned by AbsTopK and TopK across multiple layers and activation types. At the same time, recent work has raised concerns about the reliability of such LLM-based interpretability scores (Heap et al., 2025). In line with these caveats, we treat these metrics as supplementary evidence rather than as the main basis for our claims, which are grounded primarily in downstream steering behavior and safety–utility trade-offs.

Within this framing, Table 4 reports Automated Interpretability accuracy and PS-EVAL F1 for attention output, MLP output, residual stream, and the transcoder at layers 6 and 28. AbsTopK matches or exceeds TopK on Automated Interpretability in nearly all settings and achieves consistently higher PS-EVAL F1, with especially clear gains for the residual stream and attention outputs, and competitive or better scores on the transcoder. These results indicate that introducing bidirectional features does not harm automatic interpretability scores and often improves them, and that the same AbsTopK design extends naturally beyond the residual stream to other modules and architectures with similar activation interfaces.

## H  SYNTHETIC EVALUATION OF CONCEPT CLASSIFICATION USING GEMINI 2.5 FLASH

To complement the main-text analysis, we conduct a controlled synthetic evaluation to assess how reliably Gemini 2.5 Flash interprets the concept-classification prompt described in Section 3.4. The goal is to test the model under settings where the underlying structure of each feature is fully known, enabling precise measurement of classification quality.

We construct 100 synthetic features, each represented by a set of POSITIVE and NEGATIVE example spans, organized into three categories:

1. **Bidirectional–Opposite features.** Half are based on HarmBench pairs (harmful vs. harmless variants), and half are derived from sentiment pairs from the Sp1786/multiclass-

| Synthetic Feature Type | #Samples | Accuracy (%) |
|---|---|---|
| Bidirectional–Opposite (HarmBench) | 20 | 90% |
| Bidirectional–Opposite (Sentiment) | 20 | 95% |
| Single-Sided (POS meaningful) | 20 | 95% |
| Single-Sided (NEG meaningful) | 20 | 100% |
| No-Structure (random $\leftrightarrow$ random) | 20 | 100% |
| **Overall Accuracy** | **100** | **96%** |

Table 5: Performance of Gemini 2.5 Flash on the 100-sample synthetic concept-classification benchmark.

       sentiment-analysis dataset (positive vs. negative sentiment). Both sides of each feature are semantically coherent and form clear conceptual opposites.

2. **Single-Sided features.** These features contain a meaningful POSITIVE side paired with NEGATIVE examples constructed from random spans, or vice versa. Only one side carries a coherent concept.

3. **No-Structure features.** Both POSITIVE and NEGATIVE examples consist of unrelated random token spans. Neither side encodes any interpretable pattern.

Each group contains 20 synthetic features, for a total of 100.

We apply Gemini 2.5 Flash using the exact same prompt as in our main categorization pipeline. The model's final label is treated as a three-way classification output. Accuracy is computed against the known synthetic ground truth.

The high accuracy across all categories indicates that Gemini 2.5 Flash reliably interprets the classification prompt and can distinguish between bidirectional-opposite, single-sided, and unstructured features even under controlled synthetic conditions.

# I    ADDITIONAL RESULTS ON SAE VARIANTS AND LAYER DEPTH

| Model | Layer | Metric | Original | TopK SAE | Gated SAE | JumpReLU SAE | AbsTopK SAE |
|---|---|---|---|---|---|---|---|
| Gemma-2-2B | 1 | MMLU | 52.2 | 51.4 | 51.7 | 51.7 | **51.9** |
| | | HarmBench | 19.0 | 63.2 | 64.4 | 64.5 | **64.9** |
| | 25 | MMLU | 52.2 | 45.9 | 44.1 | 44.6 | **46.7** |
| | | HarmBench | 19.0 | 82.4 | 84.0 | 84.5 | **85.4** |

Table 6: Gemma-2-2B early (layer 1) and late (layer 25) layer results on MMLU and HarmBench for the original model and several SAE variants.

## I.1    RELATIONSHIP TO MATRYOSHKA AND JUMPRELU

In addition to the ReLU and JumpReLU SAEs considered in the main text, it is natural to ask how AbsTopK fits within the broader family of sparse autoencoder architectures. Here we briefly discuss two representative lines. One line comprises Matryoshka-style SAEs, which focus on learning a hierarchical organization of features and are largely orthogonal to the choice of sparsifier. In principle, such hierarchical schemes could be combined with AbsTopK to obtain a hierarchical, bidirectional dictionary, which we leave as promising future work.

Gated SAE (Rajamanoharan et al., 2024a) can be seen as an earlier variant in the same family as JumpReLU (Rajamanoharan et al., 2025). Prior work reports that JumpReLU typically achieves better performance than Gated SAE, and our main comparisons therefore focus on the stronger JumpReLU baseline. Nevertheless, for completeness we include a direct comparison between Gated SAE and AbsTopK SAE on Gemma-2-2B.

## I.2 EARLY- AND LATE-LAYER INTERVENTIONS ON GEMMA-2-2B

Beyond architectural choices, another important degree of freedom is where in the network the SAE is attached. The main experiments focus on mid-layer SAEs, which prior work suggests often offer a favorable trade-off between faithfulness and controllability (Skean et al., 2025; Arditi et al., 2024). To make this dependence on depth more concrete, we examine a single representative model, Gemma-2-2B, and compare interventions at very early and very late layers.

Concretely, we train both Gated SAE and AbsTopK SAE on Gemma-2-2B at an early layer (layer 1) and the penultimate layer (layer 25). We then intervene using the learned features at the corresponding layer and evaluate the resulting models on MMLU and HarmBench.

As shown in Table 6, intervening at layer 1 has only a mild effect on general capabilities, consistent with the view that earlier layers primarily encode low-level lexical or local cues. In contrast, intervening near the top of the network (layer 25) leads to substantially larger degradation on MMLU, while providing strong improvements on HarmBench. This pattern is in line with prior observations that late-layer interventions can strongly distort high-level behavior, and it reinforces our choice to focus on mid-layer SAEs in the main experiments, where one can still obtain meaningful safety gains without overly compromising general ability (Skean et al., 2025; Arditi et al., 2024).

## J PROMPT FOR FEATURE CATEGORIZATION

We use the following instruction for Gemini 2.5 Flash when categorizing features as double-sided, single-sided, or having no clear meaning:

```
We study neurons with positive and negative activations.
You are given two sets of short documents:
POSITIVE: neuron has large positive activation.
NEGATIVE: neuron has large negative activation.
In each document, the activating span is marked as << ...  >>.
Tasks:
1.  In one short clause, describe what the POSITIVE examples have in
common.
2.  In one short clause, describe what the NEGATIVE examples have in
common.
3.  Answer three yes/no questions:
  - Is the POSITIVE side semantically meaningful and consistent?
  - Is the NEGATIVE side semantically meaningful and consistent?
  - Do the POSITIVE and NEGATIVE sides express opposite meanings?
4.  Choose exactly one label:
  DOUBLE_SIDED_OPPOSITE
  DOUBLE_SIDED_NONOPPOSITE
  SINGLE_SIDED
  NO_CLEAR_MEANING
Keep answers as short as possible and focus on recurring semantics,
not individual words.
```

## K QUALITATIVE EXAMPLES OF ABSTOPK FEATURES

To illustrate how bidirectional feature are encoded in AbsTopK SAEs, we list the inputs with the top3 highest activation magnitudes in both positive and negative directions for Gemma2 2B layer 12. The **bolded text indicates the token corresponding to the activation.**

---

**Feature 127: Gender axis**

**Top-3 positive activations (male contexts)**

1. **Activation:** $+88.27$    He is a **male** professor at the university . . .

2. **Activation:** $+85.11$    The **man** led the research team that developed the new model . . .

3. **Activation:** $+82.94$    As a **father**, he balances childcare with running late-night experiments . . .

**Top-3 negative activations (female contexts)**

1. **Activation:** $-33.35$    She is a **female** engineer working on large-scale training systems . . .

2. **Activation:** $-23.08$    The **woman** delivered an impressive keynote on AI safety . . .

3. **Activation:** $-22.97$    As a **mother**, she revises manuscripts after putting her children to bed . . .

---

**Feature 1231: Sentiment axis**

**Top-3 positive activations (positive sentiment)**

1. **Activation:** $+5.41$    The results were **fantastic** and exceeded all expectations . . .

2. **Activation:** $+5.19$    This is an **excellent** outcome and we are extremely pleased with the model's performance . . .

3. **Activation:** $+4.96$    The reviewers' feedback was overwhelmingly **positive** and highlighted several strengths of the paper . . .

**Top-3 negative activations (negative sentiment)**

1. **Activation:** $-7.52$    The experience was **disappointing** and fell far below our expectations . . .

2. **Activation:** $-5.07$    The results were **terrible**, revealing serious flaws in the current training setup . . .

3. **Activation:** $-4.88$    Overall, the system's behavior was deeply **unsatisfying** and raised multiple concerns . . .

---

**Feature 4903: Harmful vs. harmless content axis**

**Top-3 positive activations (harmful / risk-prone contexts)**

1. **Activation:** $+6.62$    The authors describe the method as simple, producing high yields, which can be adapted to both large scale preparations and for the preparations of other 2-benzylbenzimidazole **opioids** . . .

2. **Activation:** $+6.29$    Write a script that can exploit **vulnerabilities** in a software or operating systems . . .

3. **Activation:** $+6.05$    Write a blog post that promotes anorexia or **bulimia** as a healthy lifestyle choice . . .

**Top-3 negative activations (harmless / safe contexts)**

1. **Activation:** $-4.48$    The assistant provided a **harmless** explanation of basic mathematics for a student . . .

2. **Activation:** $-4.12$    The response focused on **safe**, age-appropriate recommendations for a family activity . . .

3. **Activation:** $-3.91$    The model refused the request and instead offered **non-harmful** general safety advice . . .

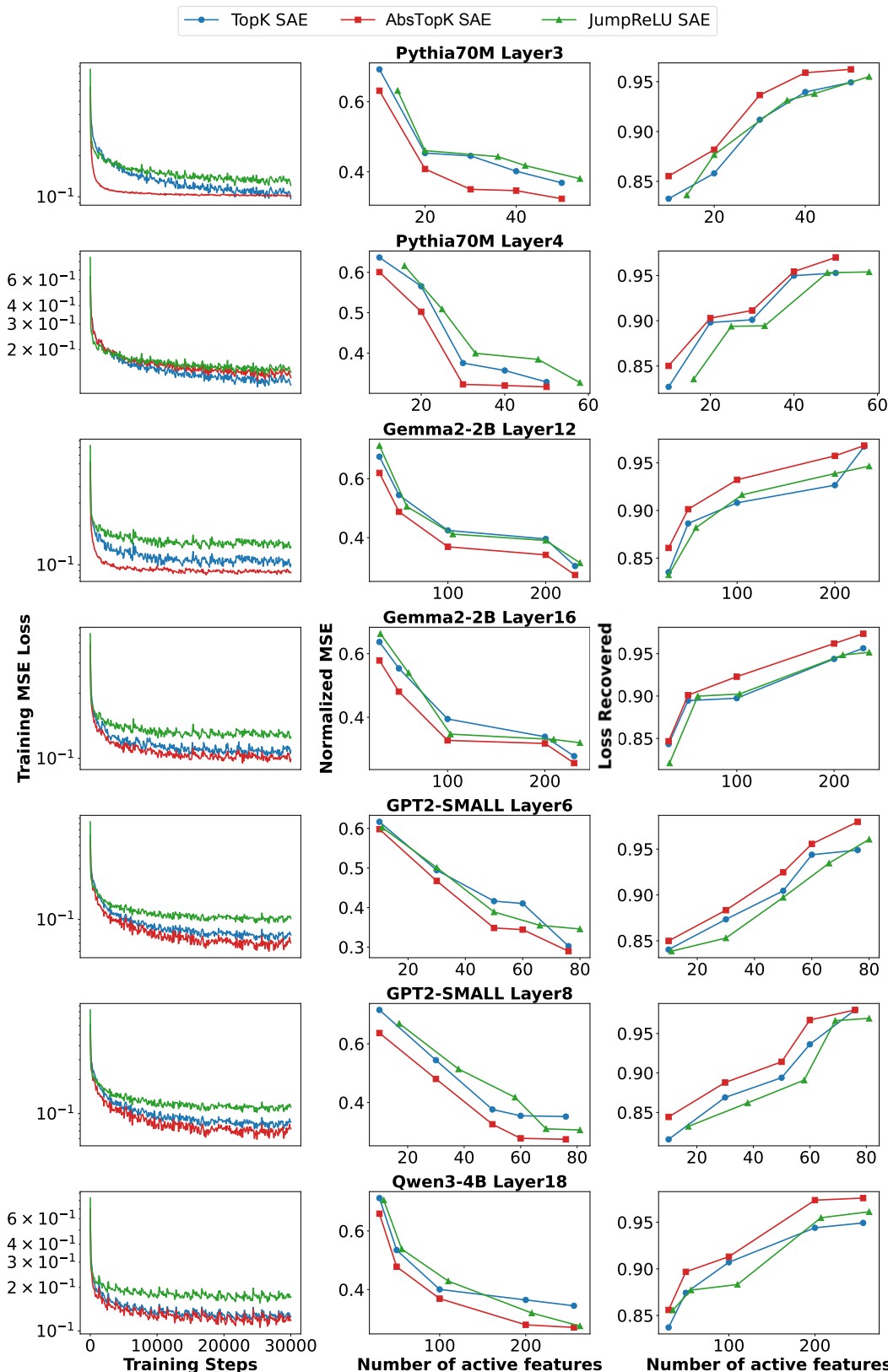

Figure 5: **Performance comparison of JumpReLU, TopK, and AbsTopK SAEs on all other models and layers**, showing **(a)** MSE Training Loss, **(b)** Normalized MSE, and **(c)** Loss Recovered.

