# OpenReview forum: "AbsTopK: Rethinking Sparse Autoencoders For Bidirectional Features"
_ICLR.cc/2026/Conference — ICLR 2026 Poster_

### Official Review · Reviewer_1nSy · 2025-10-19

**Soundness:** 3
**Presentation:** 4
**Contribution:** 3
**Rating:** 6
**Confidence:** 3

**Summary:**

The authors mathematically prove an equivalence between commonly used SAE architectures and sparse regularizers. They then use the insights gained from this connection to propose a new type of SAE which is able to encode bidirectional features as a single dictionary element. They empirically demonstrate that this new SAE architecture outperforms pre-existing SAEs on many commonly used evaluations.

**Strengths:**

1. The authors make a nontrivial connection between SAEs and sparse regularizers using proximal operators, which gives a new perspective on the differences between SAE architectures. This seems to be well-executed (though I didn’t go through the math and proof in too much detail).
2. The evaluations are quite comprehensive in terms of covering all the relevant axes of SAE quality. They demonstrate that AbsTopK are a clear improvement over JumpReLU and TopK SAEs (even though the size of the improvement is relatively small on most evaluations).
3. The SAE variant they propose is clean and elegant in how it addresses the problem of bidirectional features.

**Weaknesses:**

1. The evaluations should include more baselines. They currently include only 2: TopK and JumpReLU. They should include a handful of others (even if they don’t neatly fit into the framework introduced in section 2). ReLU, Matryoshka, and gated SAEs would all be welcome additions.
2. I want to see more empirical results showing how bidirectional features are encoded in AbsTopK SAEs. The single example in Figure 1 is nice, but it’s just a single example which isn’t sufficient to convince me that this is representative.
3. I’d like to see results from more layers. Currently as far as I can tell they only test on 2 layers in each model, and they are always somewhere roughly in the middle of the model. Include early layers and late layers would give me more confidence that the observed improvements generalize.
4. For the experiments Table 1, it would be good to see 4ish different models rather than just 2\. I agree with the reasoning for excluding the very small models here, but it would be nice to add e.g. a Llama model. I’d also weakly recommend turning table 1 into a series of plots (e.g. with MMLU on the x-axis and HarmBench on the y-axis); it would make it more easily digestible.
5. More generally in all experiments, it would be nice to include one or two larger models (e.g. 10B+ parameters).
6. Typo: the legend inside of each subfigure in Figures 2, 3, and 4 refers to “AbsoluteK”. I’m assuming this refers to AbsTopK, perhaps the authors switched from one name to the other at some point during the writing process but forgot to update the figures.

**Questions:**

See the Weaknesses section. I’d particularly recommend focusing on points 1 and 2, I see those as the main weaknesses of the paper, if you address them then I’m likely to raise my score.

---

> ### Author Response · Authors · 2025-11-23
> **Reply to Reviewer 1nSy**
>
> >W1, W3, W4, \& W5: Expanded Evaluation on Baselines, Models, Scales, and Layers
>
> We appreciate the reviewer’s feedback calling for a broader empirical evaluation, including more baselines, more layers, additional models, and larger scales. In the revision, we have expanded our experiments along all four axes.
>
> First, in terms of model diversity and scale (W4, W5), we extend the evaluation from the original two models to **four** models, now including **Llama-3.1-8B** and **Gemma-3-12B**. The addition of Gemma-3-12B directly addresses the request for a 10B+ model. As summarized in Table 1, AbsTopK consistently matches or improves upon the safety–utility trade-off across these different architectures and parameter scales.
>
> Second, to address the concern about layer coverage (W3), we move beyond mid-layer evaluations and include **early** (Layer 6) and **late** (Layer 40) layers in Gemma-3-12B, in addition to the mid layers considered on Qwen3-4B and Gemma-2-2B. Across these depths, AbsTopK continues to improve HarmBench performance while maintaining competitive MMLU accuracy, suggesting that the benefits are not confined to a narrow band of layers.
>
> Third, regarding baselines (W1), we have added a standard **ReLU SAE** baseline. On Qwen3-4B and Gemma-2-2B, where we train the full set of methods, AbsTopK maintains its advantage in the safety–utility trade-off. For the larger Llama-3.1-8B and Gemma-3-12B models, we currently include ReLU as a representative non-negative SAE baseline due to computational constraints; TopK and JumpReLU runs on these models are ongoing and will be added to the final version to further strengthen the comparison.
>
> Finally, in terms of presentation (W4), we retain Table 1 for precise numerical comparison and now explicitly report, for each SAE and model, the absolute change in MMLU and HarmBench relative to the unsteered base model. This makes comparisons across SAE types more straightforward, without relying on joint scatter plots in which the very different numerical ranges of the two metrics can be visually misleading.
>
> Overall, these new experiments show that our conclusions are robust across model families (Llama, Gemma, Qwen), model sizes (up to 12B), layer positions (early, middle, late), and multiple SAE baselines.
>
> > W2: I want to see more empirical results showing how bidirectional features are encoded in AbsTopK SAEs. The single example in Figure 1 is nice, but it’s just a single example which isn’t sufficient to convince me that this is representative.
>
> In the revision, we added both quantitative and qualitative evidence that bidirectional features are common and robust in AbsTopK SAEs.
>
> On the quantitative side, we performed an LLM-based automatic interpretation study on Gemma-2-2B with AbsTopK SAEs at layers 12 and 16 (Table 2 in Section 3.4). For each of 2000 randomly sampled features, we collected their top positive and negative activations and prompted Gemini-2.5 Flash to classify whether the feature has double-sided meaning, single-sided meaning, or no clear meaning, with a subset of double-sided features further identified as expressing opposite meanings. For AbsTopK, roughly 29--31% of features are judged double-sided and **around 20--22% explicitly encode opposite meanings**, while 56--58% are single-sided and only 11--14% lack a clear interpretation. This shows that bidirectional semantics are not rare corner cases: a substantial fraction of features systematically use their two polarities in a meaningful way.
>
> On the qualitative side, we have complemented **three additional case studies** in the Appendix H that visualize how AbsTopK SAEs encode bidirectional semantics in different domains. These examples illustrate features where the positive and negative activations correspond to opposing sentiment, gender, and safety. Together with the automatic analysis above, these additions show that the behavior is representative of a broader pattern in AbsTopK SAEs rather than an isolated example. If there are specific types of features or qualitative analyses you would be interested in, we would be happy to incorporate them in the final version.
>
> > W6: Typo
>
> Thank you for pointing this out. We will revise it in the paper.

---

> ### Comment · Reviewer_1nSy · 2025-11-24
> **Clarification**
>
> I thank the authors for their detailed response\! It’s great to see the quantitative analysis of what fraction of features follows this pattern, as well as the additional examples. I also really appreciate them including the larger models.
>
> A few follow-up questions/requests:
>
> 1. Have you tested how well Gemini 2.5 Flash can perform the classification? (E.g. generating a small synthetic dataset where you know the ground truth and asking Gemini to classify it, then reporting the F1 score)
> 2. Where can I see the new ReLU SAE results? They don’t seem to be present in the updated paper.
> 3. As I wrote in my review, I would also like to see other SAE architectures used as baselines, incl. gated and Matryoshka. I get that Matryoshka can be a bit of a hassle, but including gated SAEs should be fairly straightforward.
> 4. The additional layers are nice to see, but they still aren’t *that* early-stage or late-stage. Can you also report e.g. layer 1 and the penultimate layer? If possible, I’d love to see you take one model (possibly one of the smaller ones to make this more tractable) and report results for all of its layers, that way I can be certain that you aren’t cherry-picking the layer indices.

---

> ### Author Response · Authors · 2025-11-26
> **Reply to Clarification (1/2)**
>
> We thank the reviewer for acknowledging our responses and for providing further comments and suggestions.
>
> > As I wrote in my review, I would also like to see other SAE architectures used as baselines, incl. gated and Matryoshka. I get that Matryoshka can be a bit of a hassle, but including gated SAEs should be fairly straightforward. The additional layers are nice to see, but they still aren’t that early-stage or late-stage. Can you also report e.g. layer 1 and the penultimate layer? If possible, I’d love to see you take one model (possibly one of the smaller ones to make this more tractable) and report results for all of its layers, that way I can be certain that you aren’t cherry-picking the layer indices.
>
> **On Gated SAE.** Thank you for the further comment. In the previous response, we included ReLU SAE in Table 1 following the reviewer’s suggestion. We did not compare with Matryoshka because Matryoshka and AbsTopK are not competing or contrastive approaches, as the former focuses on hierarchical feature organization. In fact, they can be combined to learn better SAEs. We believe this may be a promising direction to explore in future work. Regarding Gated SAE [3], it is an earlier version of JumpReLU [4] proposed by the same group, and prior work has observed that JumpReLU achieves better performance. For this reason, and due to limited time and computational resources, we had not included comparisons with Gated SAE. To address the reviewer’s request for comparisons with Gated SAE as well as evaluations at both early and late layers within our resource constraints, we have now trained both Gated SAE and AbsTopK SAE on Gemma-2-2B at layer 1 and the penultimate layer.
>
> As shown in the table below, intervening at the very early layer has only a mild effect on general capabilities, consistent with the view that earlier layers primarily encode low-level cues. In contrast, intervening near the top of the network (layer 25) yields much larger degradation in MMLU. This pattern is consistent with prior observations that late-layer interventions can strongly distort high-level behavior, and it motivates our choice to focus on mid-layer SAEs in the main experiments, which provide a better trade-off between preserving general ability and improving specific behavior [1,2]. We have included this result and discussion in Appendix I in the revision and will update the results for other SAEs in the final version.
>
> | **Layer** | **Metric**   | **Original** | **Gated SAE** | **AbsTopK SAE** |
> |:--------:|:-------------|:------------:|:-------------:|:---------------:|
> | 1        | MMLU         | 52.2         | 51.7          | **51.9**        |
> | 1        | HarmBench    | 19.0         | 64.4          | **64.9**        |
> | 25       | MMLU         | 52.2         | 44.1          | **46.7**        |
> | 25       | HarmBench    | 19.0         | 84.0          | **85.4**        |
>
> *Table: Gemma-2-2B early (layer 1) and late (layer 25) layer results on MMLU and HarmBench for the original model, Gated SAE, and AbsTopK SAE.*
>
> **On “all layers.”**
> We would very much like to perform a full layer-by-layer sweep, but given our limited computational resources, training multiple SAEs (e.g., JumpReLU, TopK, AbsTopK) for every layer would require time beyond the discussion period, even for relatively small models such as Gemma-2-2B (26 layers) and Pythia-160M (12 layers). Instead, following prior work showing that mid-layer interventions are often most effective for shaping behavior without severely degrading general capabilities [1,2], we focus our main experiments on mid layers and add targeted early/late-layer checks. In the current revision, we already train SAEs on nine layers across four models, including early, mid, and late layers, which provides broad coverage and makes it unlikely that our results hinge on a single cherry-picked layer.
>
> For very small models such as Pythia-70M (6 layers), a full layer-by-layer sweep would be feasible for us, though these models exhibit weak performance on MMLU and HarmBench, as also acknowledged by the reviewer in W4 of the original comment. We plan to run these experiments and include the training curves (normalized MSE and loss recovered as in Figure 5) in a future version. In addition, if the reviewer is particularly interested in specific layers of certain models, we would be happy to run those experiments and include the corresponding results.
>
> [1] Layer by Layer: Uncovering Hidden Representations in Language Models, 2025.
> [2] Refusal in Language Models Is Mediated by a Single Direction, 2024.
> [3] Improving Dictionary Learning with Gated Sparse Autoencoders.
> [4] Jumping Ahead: Improving Reconstruction Fidelity with JumpReLU Sparse Autoencoders.

---

> > ### Author Response · Authors · 2025-11-26
> > **Reply to Clarification (2/2)**
> >
> > > Have you tested how well Gemini 2.5 Flash can perform the classification? (E.g. generating a small synthetic dataset where you know the ground truth and asking Gemini to classify it, then reporting the F1 score)
> >
> > Thanks for the great suggestion. We appreciate the reviewer's concern regarding the model's ability to classify features into the three categories. We have added a small synthetic evaluation where the ground-truth structure of each feature is fully controlled via:
> >
> > - **Bidirectional–opposite features:**
> >   Half are built from HarmBench cases (harmful vs. harmless variants), and half from the Sp1786/multiclass-sentiment-analysis-dataset (positive vs. negative sentiment). For these, both sides are semantically meaningful and form clear opposites.
> >
> > - **Single-sided features:**
> >   Half have a meaningful **POSITIVE** side paired with **NEGATIVE** examples formed by random token spans, and half are constructed symmetrically (meaningful **NEGATIVE** side, random **POSITIVE** side).
> >
> > - **No-structure features:**
> >   Both POSITIVE and NEGATIVE examples are random token spans, so neither side has a coherent concept.
> >
> > We then apply Gemini 2.5 Flash with exactly the same prompt as in our main analysis and treat the final label it outputs as a multi-class prediction. Here are the results:
> >
> > | **Synthetic Feature Type**                 | **#Samples** | **Accuracy (%)** |
> > |-------------------------------------------|--------------|------------------|
> > | Bidirectional–Opposite (HarmBench)        | 20           | 90               |
> > | Bidirectional–Opposite (Sentiment)        | 20           | 95               |
> > | Single-Sided (POS meaningful)             | 20           | 95               |
> > | Single-Sided (NEG meaningful)             | 20           | 100              |
> > | No-Structure (random ↔ random)            | 20           | 100              |
> > | **Overall Accuracy**                      | **100**      | **96**           |
> > *Table: Gemini 2.5 Flash performance on the 100-sample synthetic concept-classification benchmark.*
> >
> > This demonstrates the model's ability to classify each feature into the three categories. We have added the result and discussions in Appendix H in the updated revision.
> >
> > > Where can I see the new ReLU SAE results? They don’t seem to be present in the updated paper.
> >
> > The new ReLU SAE results are in Table 1, in the fifth column labeled “ReLU SAE” for each model/layer. We have marked it in purpule in the updated revision.

---

> > > ### Comment · Reviewer_1nSy · 2025-11-27
> > > **Response**
> > >
> > > Thank you for adding these in, I'm increasing my score

---

> > > > ### Author Response · Authors · 2025-11-28
> > > >
> > > > Thank you for responding and updating score on our work. We are glad that our rebuttal addressed concerns and we would be happy to address any further questions or suggestions you may have. We will revise our paper based on your advice.

---

### Official Review · Reviewer_mJ7S · 2025-10-29

**Soundness:** 3
**Presentation:** 3
**Contribution:** 3
**Rating:** 6
**Confidence:** 4

**Summary:**

This paper introduces a principled framework that derives sparse autoencoders (SAEs) from the proximal gradient method for sparse coding, showing that common variants like ReLU, JumpReLU, and TopK naturally emerge as single-step updates. The analysis reveals a key limitation of existing SAEs—their non-negativity constraints prevent features from capturing bidirectional semantics (e.g., male vs. female), leading to fragmented representations. To address this, the authors propose AbsTopK SAE, which applies magnitude-based hard thresholding to preserve both positive and negative activations. Experiments across multiple LLMs and interpretability tasks show that AbsTopK improves reconstruction, enhances interpretability, and enables single features to represent contrasting concepts, matching or surpassing supervised baselines.

**Strengths:**

1.	Although sparse autoencoders have emerged as a promising approach to improving the interpretability of large language models, their theoretical understanding remains limited. This paper proposes a novel framework based on a proximal perspective to analyze SAEs. The theoretical analysis is natural, rigorous, and insightful, providing meaningful guidance for future research in this area.
2.	The paper convincingly demonstrates that ignoring negative components in the latent space degrades the performance of sparse autoencoders. Replacing traditional activation functions with AbsTopK is a reasonable and well-motivated solution, supported by both solid theoretical reasoning and comprehensive empirical validation.
3.	The writing and presentation are clear, coherent, and well-structured. The theoretical and empirical sections complement each other effectively, and the overall logic flow is easy to follow. The theoretical part offers sufficient intuition without being overly mathematical. I appreciate the clarity and accessibility of the presentation.
4.	The authors provide thorough experimental verification showing that AbsTopK SAEs outperform existing variants in both reconstruction and interpretability-related tasks.

**Weaknesses:**

1.	My main concern is whether AbsTopK may compromise the monosemanticity property of sparse autoencoders. One of the most desirable characteristics of SAEs is that each dimension is activated primarily by a single concept. However, since AbsTopK activates both positive and negative top-K components, this property might be weakened. For instance, if a single feature dimension responds to both male and female concepts, it becomes difficult to isolate them semantically. In addition, the paper does not provide an evaluation of monosemanticity metrics, such as auto-interpretability scores. I would be happy to raise my evaluation if the authors can clarify or empirically address this concern.
2.	As the effectiveness of AbsTopK SAEs remains somewhat uncertain, I consider the main contribution of this paper to be its theoretical framework for understanding SAEs. Therefore, in the related work section, it would strengthen the paper to include a more explicit comparison with existing theoretical analyses of SAEs, such as [1], [2], and related studies.


[1] Chen, Siyu, et al. "Taming Polysemanticity in LLMs: Provable Feature Recovery via Sparse Autoencoders." arXiv preprint arXiv:2506.14002 (2025).
[2] Cui, Jingyi, et al. "On the Theoretical Understanding of Identifiable Sparse Autoencoders and Beyond." arXiv preprint arXiv:2506.15963 (2025).

**Questions:**

See Weaknesses.

---

> ### Author Response · Authors · 2025-11-23
> **Reply to Reviewer mJ7S**
>
> > My main concern is whether AbsTopK may compromise the monosemanticity property of sparse autoencoders...
>
> Thanks for the question. Our goal with AbsTopK is not to trade away monosemanticity for additional expressivity, but to remove a structural constraint that artificially fragments semantic axes, while still encouraging each latent to behave as a single, sparse feature.
>
> In our framework, a dimension corresponds to a decoder column and is best viewed as a semantic axis. Allowing both positive and negative activations does not mean encoding two unrelated concepts in a single neuron; rather, it permits a single axis to represent a coherent concept family with two polarities. In the male–female example, the feature remains monosemantic at the axis level: both sides consistently refer to gender, and the sign disambiguates which side of the same underlying factor is present. **What would violate monosemanticity is not having male vs. female on one axis, but, say, male vs. programming language on the same axis.** Allowing both positive and negative activations can make the learned features easier to interpret by capturing contrastive examples such as male vs. female.
>
> To empirically address this concern, we added two forms of monosemanticity evaluation in the revision. First, we performed an LLM-based automatic interpretation study on Table 2 of Section 3.4. For each of 2000 randomly sampled features, we collected top positive and negative activations and asked Gemini-2.5 Flash to judge whether the feature has double-sided meaning, single-sided meaning, or no clear meaning, and to identify the subset of double-sided features whose two polarities express opposite meanings. For AbsTopK, only 11-14% lack a clear interpretation. By contrast, for TopK 15-16% have no clear meaning. **Thus AbsTopK increases the prevalence of coherent bidirectional axes without increasing the fraction of uninterpretable features**, which directly addresses the reviewer’s concern that a single unit might mix unrelated concepts.
>
> Second, we report standard auto-interpretability metrics in the revised version. Our evaluation employs Automated Interpretability [3] to verify the semantic coherence of the learned features and PS-EVAL [4] to quantify the model's ability to disentangle polysemantic neurons into distinct, monosemantic concepts. **On Llama-3.1-8B across residual, MLP, and attention components (Table 4 in Appendix G), AbsTopK attains consistently high Automated Interpretability scores and competitive PS-EVAL performance compared to TopK SAEs.** This indicates that relaxing non-negativity does not degrade monosemanticity as measured by existing automatic interpretability metrics.
>
> Taken together, these results support the view that AbsTopK preserves monosemanticity at the axis level.
>
> [3] Language models can explain neurons in language models, 2023.
>
> [4] Rethinking evaluation of sparse autoencoders through the representation of polysemous words, ICLR2025.
>
> > As the effectiveness of AbsTopK SAEs remains somewhat uncertain, I consider the main contribution of this paper to be its theoretical framework for understanding SAEs. Therefore, in the related work section, it would strengthen the paper to include a more explicit comparison with existing theoretical analyses of SAEs, such as [1], [2], and related studies.
>
> We appreciate the reviewer's recognition of our theoretical framework and we thank for pointing out these references. While the additional evaluations introduced in response to Q1 are intended to strengthen the empirical case for AbsTopK SAEs, in the revised manuscript, we have expanded the **Related Work** section to explicitly compare our perspective with recent theoretical analyses such as [1,2]. These works focus on identifiability and study conditions under which standard non-negative SAEs can provably recover ground-truth features, typically assuming sparse, non-negative latent codes. Under these assumptions, enforcing non-negativity is theoretically well aligned with the model.
>
> Our work takes a complementary view. Motivated by the empirical observation that LLM representations often exhibit mixed-sign or bidirectional directions, we adopt a proximal-gradient perspective and show that standard SAEs implement the proximal operator of a sparsity-inducing regularizer together with an implicit non-negativity constraint. This makes clear that non-negativity is a modeling choice rather than a necessity, and leads directly to AbsTopK as the proximal operator for the same sparse coding objective without the sign constraint. Conceptually, [1,2] explain why ReLU/TopK SAEs are appropriate when underlying concepts are unipolar and non-negative, whereas our framework extends this line of reasoning to signed or bidirectional latent geometries and reduces to standard SAEs in the unipolar limit. We now highlight this relationship explicitly in the revised Related Work section.

---

> > ### Comment · Reviewer_mJ7S · 2025-11-26
> >
> > Thanks for your detailed discussions. I will keep my rating and increase my confidence.

---

> > > ### Author Response · Authors · 2025-11-26
> > >
> > > Thank you for responding and updating confidence on our work. We are glad that our rebuttal addressed concerns and we would be happy to address any further questions or suggestions you may have.

---

### Official Review · Reviewer_zJyM · 2025-10-30

**Soundness:** 1
**Presentation:** 2
**Contribution:** 1
**Rating:** 2
**Confidence:** 4

**Summary:**

This paper proposes AbsTopK SAE, a new variant of sparse autoencoder derived from the proximal gradient perspective on dictionary learning. The authors revisit the mathematical foundation of SAEs, showing that existing variants (ReLU, JumpReLU, TopK) can be unified under the proximal operator framework. From this analysis, they identify a structural limitation: current SAEs enforce non-negativity in activations, preventing single features from representing bidirectional concepts (e.g., male–female, positive–negative sentiment).
To overcome this, they introduce the AbsTopK operator, which selects the top-|k| activations by absolute magnitude, thus allowing both positive and negative activations. Experiments across several LLMs and benchmarks (probing, steering, MMLU, HarmBench) show that AbsTopK achieves better reconstruction fidelity, preserves model utility, and can encode contrasting semantics within a single feature dimension.

**Strengths:**

- Principled theoretical framing – The proximal-operator derivation provides a unifying mathematical lens for understanding existing SAE variants and justifies design differences between ReLU, JumpReLU, and TopK in a coherent way.
- Simple and reproducible modification – The AbsTopK operator is straightforward to implement and can be directly integrated into existing SAE pipelines.

**Weaknesses:**

- Lack of feature interpretability evaluation – The paper’s central claim concerns semantic bidirectionality, yet no rigorous analysis or quantitative metric is provided to assess whether positive and negative activations of a single feature correspond to semantically opposite concepts.
- Many linguistic or conceptual axes are not naturally symmetric or have no meaningful “opposite” (e.g., tree, city).
- Without systematic qualitative or quantitative validation (e.g., feature visualization, activation clustering, or concept alignment), the interpretability claim remains speculative.
- Interpretability vs. utility conflation – Improved reconstruction or downstream task performance does not necessarily imply better interpretability; this distinction should be explicitly discussed.

**Questions:**

Could you quantitatively evaluate whether positive/negative activations of a single AbsTopK feature correspond to semantically opposite text examples?
It would be useful to add qualitative visualizations (e.g., top positive vs. negative activating examples) for multiple features, not just one or two, to support the bidirectionality claim.

---

> ### Author Response · Authors · 2025-11-23
> **Reply to Reviewer zJyM**
>
> > Lack of feature interpretability evaluation – The paper’s central claim concerns semantic bidirectionality, yet no rigorous analysis or quantitative metric is provided to assess whether positive and negative activations of a single feature correspond to semantically opposite concepts.
>
> >Without systematic qualitative or quantitative validation (e.g., feature visualization, activation clustering, or concept alignment), the interpretability claim remains speculative.
>
> > Interpretability vs. utility conflation – Improved reconstruction or downstream task performance does not necessarily imply better interpretability; this distinction should be explicitly discussed.
>
> > Could you quantitatively evaluate whether positive/negative activations of a single AbsTopK feature correspond to semantically opposite text examples? It would be useful to add qualitative visualizations (e.g., top positive vs. negative activating examples) for multiple features, not just one or two, to support the bidirectionality claim.
>
> **We did not originally include automatic interpretability metrics because recent work has raised  concerns about their reliability [3].**
> Therefore, we mainly test the performance on probe and steering tasks following utility benchmarks.
> That being said, following reviewers’ suggestions, we have added experiments for interpretability metrics. We view these metrics as supplementary evidence rather than the primary basis for our claims, which remain grounded in downstream steering behavior and safety–utility trade-offs.
>
> **First, we validated that AbsTopK preserves high general interpretability standards in Appendix G.** We evaluated Llama-3.1-8B across multiple components (residual stream, MLP, and attention) using two standard metrics. Automated Interpretability [1] is used to assess semantic coherence of features, while PS-EVAL [2]  measures the recovery of sparse ground-truth concepts. As detailed in Table 4, **AbsTopK achieves consistently high Automated Interpretability scores and stable performance under PS-EVAL.** These results indicate that relaxing the non-negativity constraint does not degrade the semantic purity of the learned features and instead allows for a more flexible representation of the underlying geometry. We are currently finalizing the corresponding baselines for TopK SAEs and will include the complete comparative analysis in the final manuscript.
>
> Second, to quantify how often AbsTopK features encode semantically opposite versus unipolar concepts, we designed a dedicated LLM-based classification pipeline in Section 3.4. We randomly sampled 2,000 features from AbsTopK SAEs trained on Gemma-2-2B (Layers 12 and 16) and prompted Gemini 2.5 Flash to analyze their top positive and negative activating examples. For each feature, the LLM assigned one of three labels: bidirectional meaning, single-sided meaning, or no clear meaning. We applied the same protocol to TopK SAEs trained on the same layers. As summarized in Table 2, we find that while unipolar concepts are common, **a substantial fraction of features are effectively used to represent bidirectional concepts, and roughly 20\% explicitly encode semantic opposites.** This provides quantitative evidence that the model meaningfully exploits the signed capacity of AbsTopK when appropriate.
>
> Finally, to provide concrete intuition for these effects, we have **added three qualitative case studies** in the revised paper, as detailed in Appendix H that visualize how AbsTopK SAEs encode bidirectional semantics, highlighting features whose positive and negative activations correspond to clearly interpretable opposing concepts.
>
> [1] Language models can explain neurons in language models, 2023.
>
> [2] Rethinking evaluation of sparse autoencoders through the representation of polysemous words, ICLR2025.
>
> [3] Sparse autoencoders can interpret randomly initialized transformers, 2025.
>
> > Many linguistic or conceptual axes are not naturally symmetric or have no meaningful ``opposite'' (e.g., tree, city).
>
> We want to clarify that **we do not intend to suggest that every feature must encode a perfectly symmetric $C^{+}$ vs. $C^{-}$ axis.** Rather, our claim is about representational capacity: the model should be able to accommodate bidirectional features when the underlying concept is naturally bipolar, instead of precluding such features through non-negativity constraints. At the same time, our new formulation  does not prevent the model from learning unipolar concepts. For such features, only the positive direction may be used to sparsely represent the hidden embeddings. We have incorporated this discussion in the revision.

---

> > ### Comment · Reviewer_zJyM · 2025-11-26
> >
> > I thank you for your detailed replies. I have updated my score.

---

> > > ### Author Response · Authors · 2025-11-26
> > >
> > > Thank you for your response and acknowledgment of our responses. We are glad that our rebuttal addressed concerns and we would be happy to address any further questions or suggestions you may have.

---

### Official Review · Reviewer_EYut · 2025-10-30

**Soundness:** 3
**Presentation:** 3
**Contribution:** 2
**Rating:** 4
**Confidence:** 3

**Summary:**

This paper proposes AbsTopK SAE, a novel sparse autoencoder variant that removes the non-negativity constraint in conventional SAEs.
While existing SAEs such as ReLU, JumpReLU, and TopK enforce non-negative activations, they cannot represent bidirectional semantic axes (e.g., male–female, positive–negative sentiment), leading to feature fragmentation.

The authors rederive SAEs from a proximal gradient framework of sparse coding, revealing that the non-negativity arises from implicit regularizers.
They introduce AbsTopK, which performs hard thresholding over the largest-magnitude activations (i.e., L0 sparsity without sign restriction), enabling both positive and negative activations to coexist in a single feature.
Experiments across four LLMs (GPT2-Small, Pythia-70M, Gemma-2B, Qwen-4B) and seven probing/steering benchmarks show that AbsTopK improves reconstruction fidelity, interpretability, and bidirectional concept encoding, matching or even surpassing the supervised Difference-in-Mean baseline.

**Strengths:**

- AbsTopK consistently outperforms TopK and JumpReLU across reconstruction, probing, and steering tasks.

- Derivation from the proximal gradient method grounds the design of SAEs in dictionary-learning theory, showing why prior variants enforce non-negativity.

- Both theoretical and empirical validations are presented coherently.

**Weaknesses:**

- The paper assumes that features should be bidirectional (e.g., male-female, positive-negative). However, not all concepts have clearly defined opposites. For such unipolar or abstract concepts (e.g., syntax awareness, topic consistency), the interpretation of negative activations remains ambiguous. It is unclear whether these activations correspond to an absence of a feature, an opposing property, or noise.

- While the experiments use standardized benchmarks such as SAEBench, the paper would benefit from direct analyses showing how AbsTopK changes the feature space compared to the standard SAEs.

- The evaluation focuses mainly on a single activation type (residual stream) and a limited set of layers. Since SAE behavior often depends strongly on layer position and activation type (Attention, MLP, Residual Stream) [1], extending experiments across multiple layers and activation sources would make the evidence more comprehensive and convincing.

- In addition, the results are promising, but the paper does not report variability across random seeds or runs. Providing standard errors or confidence intervals would strengthen the reliability of empirical conclusions.

[1] Rethinking evaluation of sparse autoencoders through the representation of polysemous words, ICLR2025

**Questions:**

- Several recent studies challenge the Linear Representation Hypothesis [1,2]. How does the proposed framework relate to or address these counterarguments, or do you have any discussion?

- Could the AbsTopK approach be extended or adapted to Transcoder[3] or CrossCoder[4] architectures?

[1] Interpreting Neural Networks through the Polytope Lens.
[2] Not All Language Model Features Are One-Dimensionally Linear, ICLR2025.
[3] Transcoders Find Interpretable LLM Feature Circuits, Neurips2024.
[4] Sparse Crosscoders for Cross-Layer Features and Model Diffing.

---

> ### Author Response · Authors · 2025-11-23
> **Reply to Reviewer EYut (1/2)**
>
> > The paper assumes that features should be bidirectional (e.g., male-female, positive-negative). However, not all concepts have clearly defined opposites. For such unipolar or abstract concepts (e.g., syntax awareness, topic consistency), the interpretation of negative activations remains ambiguous. It is unclear whether these activations correspond to an absence of a feature, an opposing property, or noise.
>
> Thanks for the comments. **We want to clarify that we do not intend to suggest that every feature must encode a perfectly symmetric $C^{+}$ vs. $C^{-}$ axis.** Rather, our claim is about representational capacity: the model should be able to accommodate bidirectional features when the underlying concept is naturally bipolar, instead of precluding such features through non-negativity constraints. At the same time, our new formulation  does not prevent the model from learning unipolar concepts. For such features, only the positive direction may be used to sparsely represent the hidden embeddings. We have incorporated this discussion in the revision.
>
> To clarify how negative activations behave empirically, we performed **an LLM-based automatic interpretation analysis aimed at distinguishing bidirectional concepts from unipolar ones in Section 3.4.** We randomly sampled 2000 features, collected top positive and negative activations, and prompted Gemini-2.5 Flash to classify each feature as double-sided meaning, single-sided meaning, and no clear meaning.
>
> The results in Table 2 both corroborate the reviewer's point that many features are effectively unipolar with another side's activations behaving as absence or weak signal, and show that a non-trivial fraction are genuinely bidirectional. Overall, this supports AbsTopK does not force all features to become symmetric axes; it simply enables bidirectional features when they are supported by the data, while remaining compatible with unipolar and abstract concepts. We will incorporate this discussion and the corresponding experiments in the revised version of the paper.
>
> > While the experiments use standardized benchmarks such as SAEBench, the paper would benefit from direct analyses showing how AbsTopK changes the feature space compared to the standard SAEs.
>
> Conceptually, our proximal-gradient view illustrates this change: the only structural difference between AbsTopK and TopK lies in the removal of the non-negativity constraint in the activation update. Geometrically, this enlarges the feasible code space from the nonnegative orthant to a signed sparse set, so that a single AbsTopK feature can span both directions along a semantic axis (e.g., male--female). We add Figure 2 in the revised version of the paper to directly illustrate this one. In contrast, a nonnegative TopK SAE must represent such a bipolar axis using two separate features, fragmenting variation across distinct coordinates. Thus, AbsTopK primarily unifies pairs of opposite or complementary TopK features into a single bidirectional direction, rather than introducing qualitatively new types of directions. Additionally, we provide concrete examples of positive and negative activations to visually demonstrate this unification mechanism in the revised version of the paper.
>
> There are many possible quantitative notions of ``feature space change''. Because the current comment is somewhat open-ended, we would be grateful for clarification on which notion the reviewer has in mind. We would be happy to incorporate the corresponding geometric analysis in the revised version of the paper.
>
> > The evaluation focuses mainly on a single activation type (residual stream) and a limited set of layers....
>
> We already test 8 different layers across four models for SAEBench. In terms of activation sources, we mainly focus on residual stream because this is the most commonly used. Following reviewer's suggestion, we now have conducted additional experiments on Attention, MLP, transcode, and using Automated Interpretability [5] to verify the semantic coherence of the learned features and PS-EVAL [6] to quantify the model's ability to disentangle polysemantic neurons into distinct, monosemantic concepts.
>
> **Table 4 in Appendix G demonstrates that on components with distinct distributions, AbsTopK consistently achieves high Automated Interpretability scores ($>0.80$) and stable PS-EVAL F1 scores.** This confirms that the benefits of the AbsTopK formulation are not limited to specific residual stream properties but generalize effectively to other architectural components. We are currently finalizing the corresponding baselines for TopK SAEs and will include the complete comparative analysis in the final manuscript.
>
> [5] Language models can explain neurons in language models, 2023.
>
> [6] Rethinking evaluation of sparse autoencoders through the representation of polysemous words, ICLR2025.

---

> ### Author Response · Authors · 2025-11-23
> **Reply to Reviewer EYut (2/2)**
>
> >In addition, the results are promising, but the paper does not report variability across random seeds or runs. Providing standard errors or confidence intervals would strengthen the reliability of empirical conclusions.
>
> To ensure the reliability of our empirical claims, **we re-ran our SAEBench experiments across 5 independent random seeds (40--44).** These results, visualized in Figure 4, demonstrate that the performance gains of AbsTopK are statistically stable and robust to initialization variability
>
> >Several recent studies challenge the Linear Representation Hypothesis [1,2]. How does the proposed framework relate to or address these counterarguments, or do you have any discussion?
>
> Our work does not rely on the strongest form of LRH (``all concepts are one-dimensional and globally linear''). Instead, we rely on the weaker assumption in Eq. (2): locally, hidden states can be approximated as a sparse linear combination of latent directions. As argued in [2], SAEs can in fact be used to recover multi-dimensional features, and [1] shows that neuron-level features are often better described as lying in low-dimensional linear subspaces.
>
> **From this perspective, the concerns raised in [1,2] are orthogonal to how we invoke the linear representation hypothesis.** Those works focus on the geometry of features and emphasize that some concepts span low-dimensional subspaces, which can be captured either by multiple related neurons [1] or by a small set of SAE features [2]. Our use of LRH in Eq. (2) is compatible with this view: we assume only a local, sparse linear approximation of hidden states, not that every concept is strictly one-dimensional or globally linear. In this sense, our application of LRH does not conflict with the observations in [1,2]; rather, it operates at a different level of abstraction and remains fully consistent with their subspace-based perspective.
>
> > Could the AbsTopK approach be extended or adapted to Transcoder[3] or CrossCoder[4] architectures?
>
> Thanks for the question. To demonstrate this, we trained AbsTopK Transcoders on Llama-3.1-8B. As shown in Table 4, the **AbsTopK Transcoder successfully learns interpretable features with performance comparable to residual stream SAEs**. This confirms that our framework generalizes naturally to architectures like Transcoders without loss of interpretability.
>
> While we expect similar benefits for CrossCoders given their shared sparse-coding principles, we leave the specific implementation and evaluation on CrossCoder architectures to future work.

---

> > ### Comment · Reviewer_EYut · 2025-11-25
> >
> > Thank you for the detailed and careful responses. Your clarifications, additional experiments, and expanded analyses addressed most of my concerns.
> >
> > Given these improvements, I am increasing my score from 4 to 6.
> > I look forward to the revised version.

---

> > > ### Author Response · Authors · 2025-11-25
> > >
> > > Thank you for the positive feedback and for updating your score. We appreciate your time and look forward to further improving the paper.

---

### Comment · Area_Chair_GcDq · 2025-11-25
**Please discuss**

Several reviewers have not responded to the authors' rebuttals. Please read and respond to them. Have the rebuttals addressed your concerns or clarified anything?

---

### Author Response · Authors · 2025-12-04
**Overall Summary**

We sincerely regret the recent incident and  appreciation for the extra effort it may have required from the AC. We would like to thank the reviewers for providing constructive feedback to our work.  Below, we provide a summary of our discussion.

All reviewers acknowledge our contributions of using a unified framework using proximal operator to analyze SAEs, identifying the limitations of current SAEs, and our approach (quotes "clean and elegant", "simple and reproducible modification") in addressing the problem of bidirectional features. Below, we summarize the main three concerns and our responses. In the revision, we have expanded the corresponding parts of the main paper, highlighted in purple, to reflect these additions. Further details are given in the individual rebuttals.

- *Whether all the features are assumed as bidirectional.*

  We have clarified in Section 2 and added a new Figure 2 to state that our approach does **not** assume all features are bidirectional. Instead, it can handle both bidirectional features and unipolar concepts.

- *Lack of interpretability metrics.*

  In the original submission, we primarily evaluated performance on probing and steering tasks using recently developed benchmarks for SAEs. In the revision, we provide additional interpretability evidence based on analyses of semantic axes (Section 3.4 and Appendix H) and automatic feature evaluation with AutoInterp and PS-EVAL (Appendix G).

- *A broader empirical evaluation of our proposed SAE across more models and layers.*

  We have added experiments on Llama-3.1-8B and Gemma-3-12B, described in Section 3.3 of the main paper, to demonstrate that our findings are not driven by cherry-picked cases.


All reviewers responded to our rebuttals before the rollback. Three of the four reviewers increased their overall scores, and one reviewer increased their confidence. As a result, there is now consensus among the reviewers, and no reviewer has indicated any remaining major concerns. Reviewer mJ7S (Nov. 26) maintained their score of 6 and increased their confidence, supporting acceptance. Reviewer EYut (Nov. 25) raised their score from 4 to 6, noting that our responses addressed most of their concerns. Reviewer zJyM (Nov. 25) raised their score from 2 to 4. Reviewer 1nSy (Nov. 27) engaged in a very constructive discussion (which we greatly appreciate) and increased their score from 6 to 8.

---

### Meta-Review · Area_Chair_5Nmc · 2025-12-13

**Summary:**

Reviewers consistently recognized the paper’s core contributions. Initial concerns focused on interpretability, assumptions, and robustness of evaluation. Through extensive rebuttal and revisions, the authors substantially expanded empirical validation, added quantitative and qualitative interpretability analyses, evaluated additional models and layers, and clarified conceptual claims.

**Reviewer Concerns:**

Addressed

- Assumption was clarified
- The authors emphasized representational capacity rather than forcing symmetry
- Demonstrated empirically that AbsTopK supports both unipolar and bidirectional features
- Lack of interpretability evaluation was addressed with multiple additions
- Limited empirical scope was substantially expanded to include more models, more layers, additional activation sources, and additional architectures
- Concerns about robustness were addressed
- Monosemanticity concerns were directly tackled with both conceptual clarification and empirical metrics
- Requests for additional baselines were partially addressed

Remaining

- Some reviewers noted that further baselines and exhaustive layer-by-layer sweeps would still be desirable
- The gains in downstream metrics, while consistent, were sometimes described as modest

**Reviewer Scores:**

- Reviewer EYut: Increased score from 4 to 6 after additional interpretability analyses, broader experiments, and robustness checks. Would likely maintain this higher score after full discussion.
- Reviewer zJyM: Initially very critical, but updated their score upward after quantitative and qualitative interpretability evidence was added. Likely improved to at least borderline-positive.
- Reviewer mJ7S: Maintained a score of 6 and increased confidence after concerns about monosemanticity and related work were addressed.
- Reviewer 1nSy: Increased score after expanded baselines, larger models, broader layer coverage, and strong bidirectionality analyses. Ended clearly supportive.

---

### Decision · Program_Chairs · 2026-01-26

Accept (Poster)